# Myonectin protects against skeletal muscle dysfunction in male mice through activation of AMPK/PGC1α pathway

Yuta Ozaki[1], Koji Ohashi [2] ✉, Naoya Otaka[1], Hiroshi Kawanishi[1], Tomonobu Takikawa[1], Lixin Fang[1], Kunihiko Takahara[1], Minako Tatsumi[2], Sohta Ishihama[1], Mikito Takefuji [1], Katsuhiro Kato [1], Yuuki Shimizu [1], Yasuko K. Bando [1], Aiko Inoue[3], Masafumi Kuzuya[3,4], Shinji Miura [5], Toyoaki Murohara [1] & Noriyuki Ouchi [2] ✉

To maintain and restore skeletal muscle mass and function is essential for healthy aging. We have found that myonectin acts as a cardioprotective myokine. Here, we investigate the effect of myonectin on skeletal muscle atrophy in various male mouse models of muscle dysfunction. Disruption of myonectin exacerbates skeletal muscle atrophy in age-associated, sciatic denervation-induced or dexamethasone (DEX)-induced muscle atrophy models. Myonectin deficiency also contributes to exacerbated mitochondrial dysfunction and reduces expression of mitochondrial biogenesis-associated genes including PGC1α in denervated muscle. Myonectin supplementation attenuates denervation-induced muscle atrophy via activation of AMPK. Myonectin also reverses DEX-induced atrophy of cultured myotubes through the AMPK/PGC1α signaling. Furthermore, myonectin treatment suppresses muscle atrophy in senescence-accelerated mouse prone (SAMP) 8 mouse model of accelerated aging or mdx mouse model of Duchenne muscular dystrophy. These data indicate that myonectin can ameliorate skeletal muscle dysfunction through AMPK/PGC1α-dependent mechanisms, suggesting that myonectin could represent a therapeutic target of muscle atrophy.

Discrepancy between average life expectancy and healthy life expectancy is a serious social problem in aged societies worldwide. Age-associated loss of muscle mass and function, also known as sarcopenia, is one of the determinant factors for physical disability, thereby leading to adverse outcomes including poor quality of life and death[1]. Exercise training results in enhancement of muscle mass and function, and provides a benefit to sarcopenia. However, disabilities caused by age-associated complications including stroke, bone fracture and cancer-associated cachexia make exercise training impractical or inefficient among elderly patients. Thus, the development of therapeutic approaches to maintain and restore skeletal muscle mass and function could be indispensable for healthy aging.

Myonectin, also known as C1q/TNF-related protein 15/erythroferrone, acts as a muscle-derived secreted factor, also referred to as myokine, which is abundantly expressed in skeletal muscle tissue[2,3]. Myonectin has been reported to modulate fatty acid metabolism, iron

[1]Department of Cardiology, Nagoya University Graduate School of Medicine, Nagoya, Japan. [2]Department of Molecular Medicine and Cardiology, Nagoya University Graduate School of Medicine, Nagoya, Japan. [3]Institute of Innovation for Future Society, Nagoya University Graduate School of Medicine, Nagoya, Japan. [4]Department of Community Healthcare & Geriatrics, Nagoya University Graduate School of Medicine, Nagoya, Japan. [5]Laboratory of Nutritional Biochemistry, Graduate School of Nutritional and Environmental Sciences, University of Shizuoka, Shizuoka, Japan.
✉e-mail: ohashik@med.nagoya-u.ac.jp; nouchi@med.nagoya-u.ac.jp

metabolism, osteoblast and osteoclast differentiation and adipogenesis[4–8]. Recently we have reported that myonectin is an exercise-induced myokine which protects the heart from ischemia-reperfusion injury[9]. These findings indicated that myonectin affects nearby or remote organs in an endocrine manner to maintain whole body homeostasis. However, the impact of myonectin on skeletal muscle function and disease in an autocrine manner has not been clarified. Here, we investigated whether myonectin modulates skeletal muscle function in various mouse models of muscle dysfunction including age-related sarcopenia.

## Results

### Myonectin deficiency accelerates muscle atrophy and weakness in male aged mice

We investigated whether muscle myonectin expression is modulated by aging process. The mRNA and protein levels of myonectin (*Fam132b*) in soleus and gastrocnemius muscle tissues were significantly lower in 80-week-old (aged) WT mice than in 20-week-old (young) WT mice (Fig. 1a and Supplementary Fig. 1). To investigate whether myonectin contributes to the skeletal muscle mass and function in aged mice, muscle weight and strength in young and aged myonectin-knockout (KO) mice were evaluated. The weights of gastrocnemius and soleus muscle tissues divided by body weights were significantly lower in aged myonectin-KO mice than in aged WT mice, whereas there were no significant differences in muscle weights between young WT and young myonectin-KO mice (Fig. 1b). Aged myonectin-KO mice exhibited increased body weight compared with aged WT mice, while body weight did not differ between young WT and young myonectin-KO mice (Supplementary Fig. 2a). Gastrocnemius muscle weight was significantly lower in aged myonectin-KO mice than in aged-WT mice (Supplementary Fig. 2a). Soleus muscle weight seemed to be lower in aged myonectin-KO mice than in aged WT mice, but this was not statistically significant. There were no significant differences in gastrocnemius muscle and soleus muscle weights between young WT and young myonectin-KO mice.

Consistently, mean cross-sectional area (CSA) of gastrocnemius muscle tissues was significantly smaller in aged myonectin-KO mice than in aged WT mice, whereas there were no differences in CSA of gastrocnemius muscle tissues between young WT and young myonectin-KO mice (Fig. 1c). Consistently, aged myonectin-KO mice showed the smaller size distribution of gastrocnemius muscle CSA compared with aged WT mice (Fig. 1c). In addition, mean CSA of type II muscle fibers in gastrocnemius muscle was significantly smaller in aged myonectin-KO mice than in aged WT mice (Supplementary Fig. 3a, b). There were no differences in mean CSA of type I muscle fibers in gastrocnemius muscle between aged WT and aged myonectin-KO mice (Supplementary Fig. 3a, b). Aged myonectin-KO mice exhibited the increased frequency of smaller type II fiber CSA compared with aged WT mice, whereas the distribution of type I fiber cross-sectional area did not differ between aged WT and aged myonectin-KO mice (Supplementary Fig. 3b). Aged WT mice had the increased frequency of type I fibers in gastrocnemius muscle compared with young WT mice, and aged myonectin-KO mice exhibited the reduced frequency of type I fibers compared with aged WT mice (Supplementary Fig. 3c).

Furthermore, aged myonectin-KO mice showed the significant reduction of maximal force of grip strength in 4 limbs and fore 2 limbs, which was normalized by body weight, as compared with aged WT mice (Fig. 1d). Aged myonectin-KO mice also had the significant reduction of non-normalized maximal force of grip strength in 4 limbs compared with aged WT mice (Supplementary Fig. 2b). Non-normalized maximal force of grip strength in fore 2 limbs seemed to be lower in aged myonectin-KO mice than in aged WT mice, but this was not statistically significant. The maximal force of grip strength in 4 limbs and fore 2 limbs, which was normalized by muscle weight did not differ between aged WT and aged myonectin-KO mice. There were no

significant differences in maximal force of grip strength in 4 limbs and fore 2 limbs, which was non-normalized or normalized by body weight or muscle weight, between young WT and young myonectin-KO mice. In voluntary wheel running test, aged myonectin-KO mice exhibited remarkable reduction of average number of rotations compared with aged WT mice (Fig. 1e). Thus, myonectin reduction could promote muscle atrophy and dysfunction in aged mice.

### Myonectin deficiency exacerbates muscle atrophy induced by sciatic nerve denervation or steroid administration

To further investigate the roles of myonectin in muscle atrophy, myonectin-KO and WT mice were subjected to denervation-induced or dexamethasone (DEX)-induced muscle atrophy. Myonectin protein levels in gastrocnemius muscle tissues were significantly reduced in denervation-operated and DEX-treated WT mice compared with sham WT mice (Supplementary Fig. 4a). At 5 days after sciatic nerve denervation, the weights of denervated gastrocnemius and soleus muscle tissues normalized by body weights were significantly lower in myonectin-KO mice than in WT mice (Fig. 2a). There were no significant differences in body weight, denervated gastrocnemius muscle weight and denervated soleus muscle weight between WT and myonectin-KO mice (Supplementary Fig. 2c). There were no significant differences in the weights of non-denervated gastrocnemius and soleus muscle tissues, which was non-normalized or normalized by body weights between WT and myonectin-KO mice (Fig. 2a and Supplementary Fig. 2c).

Mean CSA of denervated gastrocnemius and soleus muscle tissues was significantly smaller in myonectin-KO mice than in WT mice, whereas there were no differences in CSA of non-denervated gastrocnemius and soleus muscle between WT and myonectin-KO mice (Fig. 2b and Supplementary Fig. 4b). Myonectin-KO mice had the smaller size distribution of CSA of denervated gastrocnemius and soleus muscle compared with WT mice (Fig. 2b, Supplementary Fig. 4b).

Mean CSA of type II muscle fibers in denervated gastrocnemius and soleus muscle was significantly smaller in myonectin-KO mice than in WT mice, whereas no differences were observed in mean CSA of type I muscle fibers in denervated muscle between WT and myonectin-KO mice (Supplementary Figs. 5a, b, and 6a, b). Myonectin-KO mice showed the increased frequency of smaller type II fiber CSA of denervated gastrocnemius and soleus muscle compared with WT mice, whereas the distribution of type I fiber CSA in denervated muscle did not differ between WT and myonectin-KO mice (Supplementary Figs. 5b and 6b). There were no differences in the frequencies of type I fibers in gastrocnemius and soleus muscle after denervation between WT and myonectin-KO mice (Supplementary Figs. 5c and 6c). Little fibrosis was observed in non-denervated or denervated gastrocnemius muscle in WT and myonectin-KO mice, and no differences were observed in the area of interstitial fibrosis between WT and myonectin-KO mice (Supplementary Fig. 7).

Furthermore, at 14 days after continuous administration of DEX, weights of gastrocnemius and soleus muscle tissues normalized by body weights compared to WT mice (Fig. 2c). Body weight did not differ between DEX-treated WT and myonectin-KO mice (Supplementary Fig. 2d). The weights of gastrocnemius and soleus muscle tissues seemed to be lower in DEX-treated myonectin-KO mice than in DEX-treated WT mice, but this was not statistically significant. There were no significant differences in body weight, gastrocnemius muscle weight and soleus muscle weight between sham-treated WT and myonectin-KO mice (Supplementary Fig. 2d).

Mean CSA of gastrocnemius muscle were significantly reduced in myonectin-KO mice after DEX treatment compared to WT mice (Fig. 2d). Myonectin-KO mice had the smaller size distribution of CSA of gastrocnemius muscle after DEX treatment compared with WT mice (Fig. 2d). Mean CSA of type II muscle fibers in gastrocnemius muscle

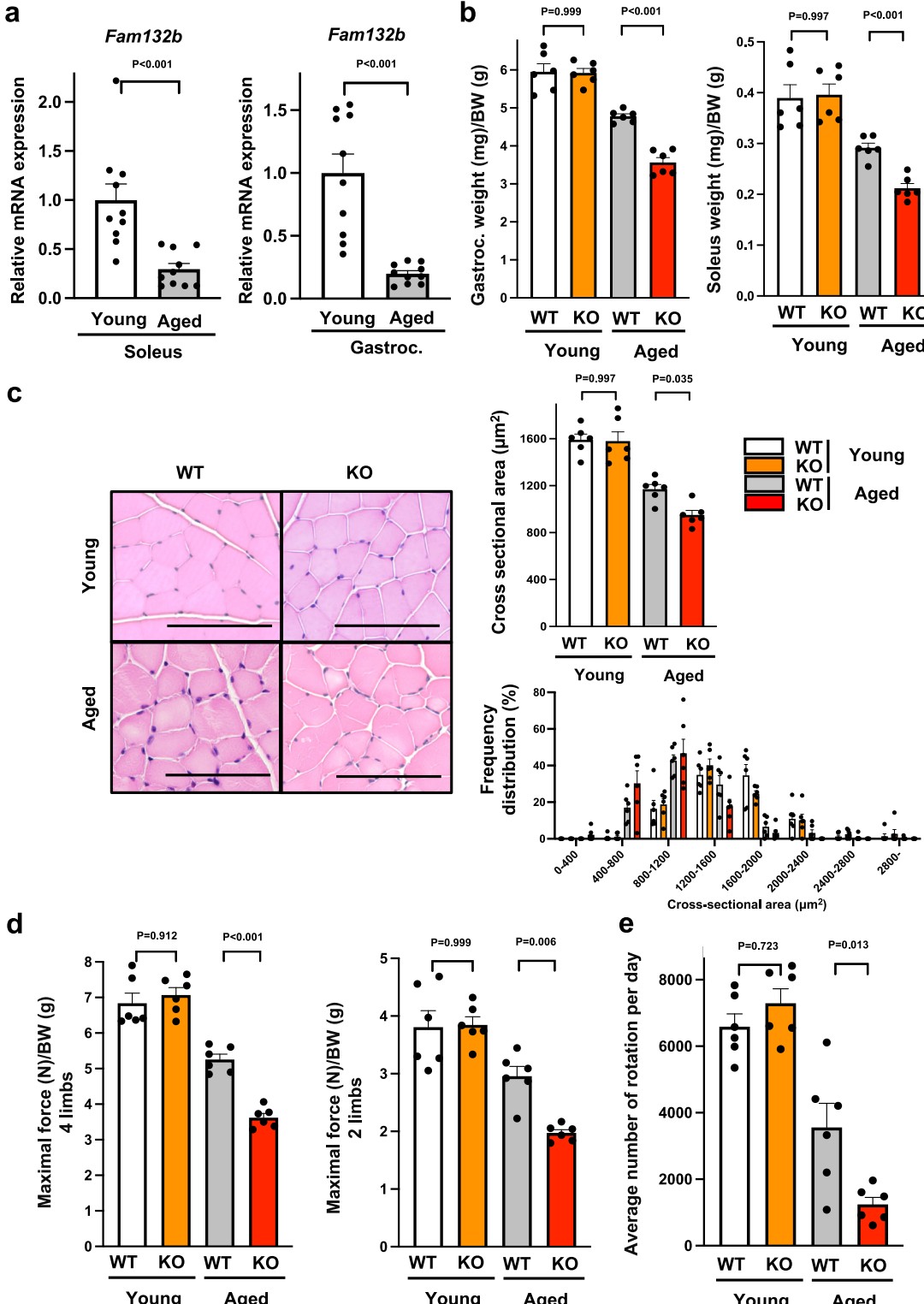

**Fig. 1 | Myonectin deficiency promotes muscle atrophy and weakness in aged mice. a** The mRNA levels of myonectin (*Fam132b*) in soleus (*N* = 9 in each group) and gastrocnemius (*N* = 10 in each group) muscles of 20-week-old young and 80-week-old aged WT mice. **b**–**e** Muscle mass and strength were assessed in 20-week-old young WT mice, 20-week-old young myonectin-KO mice, 80-week-old aged WT mice and 80-week-old aged myonectin-KO mice. **b** The ratio of gastrocnemius or soleus muscle weight to body weight is shown. *N* = 6 in each group. **c** Left panels show representative cross sectional images of gastrocnemius muscle. Scale bars show 100 μm. Right panels show quantification of mean cross sectional area (CSA) and CSA distribution. *N* = 6 in each group. **d** Maximal grip strength in 4 limbs or fore 2 limbs which is normalized by body weight is shown. *N* = 6 in each group. **e** Average number of rotation per day measured by voluntary wheel running test. *N* = 6 in each group. Data are presented as means ± SEM. Two-tailed unpaired Student's *t*-test for **a**. One-way ANOVA with a post-hoc analysis for **b**–**e** were performed. Gastroc. gastrocnemius muscle, BW body weight.

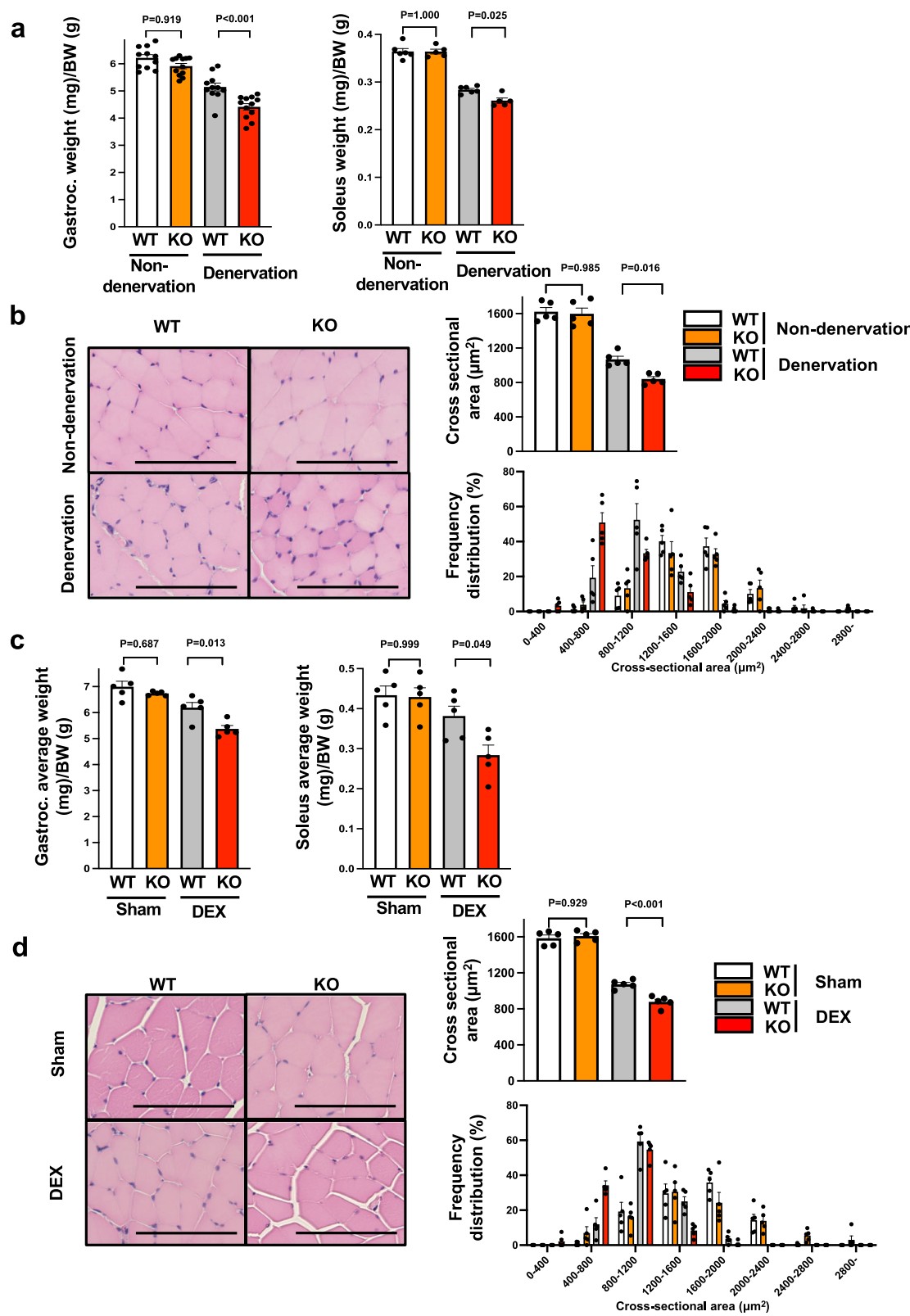

was significantly smaller in DEX-treated myonectin-KO mice than in DEX-treated WT mice, whereas there were no differences in mean CSA of type I muscle fibers between DEX-treated WT mice and DEX-treated myonectin-KO mice (Supplementary Fig. 8a, b). The increased frequency of smaller type II fiber CSA of gastrocnemius muscle was observed in DEX-treated myonectin-KO mice compared with DEX-treated WT mice, whereas the distribution of type I fiber CSA in

gastrocnemius muscle did not differ between DEX-treated WT and DEX-treated myonectin-KO mice (Supplementary Fig. 8b). No differences in the percentages of type I fibers in gastrocnemius muscle after DEX treatment between WT and myonectin-KO mice (Supplementary Fig. 8c).

Maximal force of grip strength in 4 limbs and fore 2 limbs, which was non-normalized or normalized by body weight was significantly

**Fig. 2 | Myonectin deficiency exacerbates muscle atrophy induced by sciatic nerve denervation or steroid administration. a**, **b** WT and myonectin-KO mice at the age of 8–10 weeks were subjected to sciatic denervation-induced muscle atrophy operation. At 5 days after sciatic nerve denervation, the denervated and non-denervated (contralateral) muscles were used for analyses. **a** Gastrocnemius (*n* = 11 mice per group) or soleus (*N* = 6 in non-denervated WT, non-denervated KO and denervated WT mice, *N* = 5 in denervated KO mice) muscle weight/body weight ratio is shown. **b** Left panels show representative cross sectional images of gastrocnemius muscle fibers. Scale bars show 100 μm. Right panels show quantitative analyses of mean cross sectional area (CSA) and CSA distribution of gastrocnemius muscle. *N* = 5 in each group. **c**, **d** At 14 days after continuous administration of dexamethasone (DEX) or vehicle (Sham), WT and myonectin-KO mice were sacrificed for analysis. **c** The ratio of gastrocnemius or soleus muscle weight to body weight is shown. *N* = 5 in each group. **d** Left panels show representative cross sectional images of gastrocnemius muscle fibers. Scale bars show 100 μm. Right panels show quantitative analyses of mean CSA and CSA distribution of gastrocnemius muscle. *N* = 5 in each group. Data are presented as means ± SEM. One-way ANOVA with a post-hoc analysis for **a**–**d** was performed. Gastroc. gastrocnemius muscle, BW body weight, DEX dexamethasone.

---

reduced in DEX-treated myonectin-KO mice compared with DEX-treated WT mice (Supplementary Fig. 2e). Maximal force of grip strength in 4 limbs, but not, in fore 2 limbs, which was normalized by muscle weight, was significantly reduced in DEX-treated myonectin-KO mice compared with DEX-treated WT mice. These results indicate that myonectin could modulate skeletal muscle atrophy in response to multiple pathological stimuli.

### Myonectin deficiency downregulates AMPK/PGC1α signals in denervation-induced atrophic skeletal muscle

To examine the molecular mechanism by which myonectin deficiency promotes muscle atrophy, we comprehensively analyzed the mRNA expression in denervated gastrocnemius of WT and myonectin-KO mice by RNA-Seq. Differential gene expression analysis with FDR-adjusted *p*-value < 0.05 and an absolute log2 fold change > 0.5 identified 1,399 upregulated and 573 downregulated genes in myonectin-KO mice compared with WT mice (Fig. 3a). Kyoto Encyclopedia of Genes and Genomes (KEGG) pathway enrichment analysis revealed that myonectin-dependent gene changes were associated with Pathway in cancer, Proteoglycans in cancer, Focal adhesion, Adipocytokine signaling pathway, AMP-activated protein kinase (AMPK) signaling pathway and Apelin signaling pathway (Fig. 3a). Among these differentially regulated gene sets, AMPK signaling pathway is known to regulate skeletal muscle function. Because PGC1α is a crucial downstream target of AMPK which prevents muscle dysfunction in muscle atrophy models[10–13], the expression levels of PGC1α are evaluated in denervated and non-denervated gastrocnemius muscle of WT and myonectin-KO mice by quantitative real time PCR methods. The mRNA levels of *Pgc1α* in denervated gastrocnemius muscles were significantly reduced in myonectin-KO mice compared with WT mice (Fig. 3b). Notably, the expression levels of *Pgc1α4*, which specifically relates to muscle hypertrophy[14], was also downregulated in myonectin-KO mice compared with WT mice (Fig. 3b). Likewise, protein levels of PGC1α and PGC1α4 in denervated gastrocnemius muscle were reduced in myonectin-KO mice compared with WT mice (Fig. 3c). In addition, phosphorylation levels of AMPK were reduced in denervated gastrocnemius muscle in myonectin-KO mice compared with WT mice (Fig. 3c). Furthermore, protein levels of insulin-like growth factor (IGF1), which is upregulated by PGC1α4[14], were reduced in denervated gastrocnemius muscle in myonectin-KO mice compared with WT mice (Fig. 3c). By contrast, the expression levels of ubiquitin ligase related genes including F-box only protein (*Fbxo*) 32 and tripartite motif-containing (*Trim*) 63, myostatin (*Mstn*) and myogenic genes including *Myod1* and *Myog* in denervated muscles were not different between WT and myonectin-KO mice (Fig. 3d). These results indicate that myonectin deficiency could exacerbate muscle atrophy due to the downregulation of AMPK/PGC1α pathways.

### Myonectin deficiency exacerbates mitochondrial dysfunction in denervated muscle

Because PGC1α is a key regulator of mitochondrial biogenesis[10], we investigated whether myonectin deficiency contributes to mitochondrial dysfunction in response to muscle atrophy. The expression levels

of mitochondrial biogenesis related genes including *Tfam*, *Sirt1*, *Nrf1* and *Nfe2l2* in denervated muscle were significantly lower in myonectin-KO mice compared with WT mice (Fig. 4a).

We also evaluated morphological change of mitochondria during muscle atrophy process by using transmission electronical microscopy. The number of mitochondria in denervated muscle tissues was significantly smaller in myonectin-KO mice than in WT mice (Fig. 4b). In addition, the percentage of altered mitochondria, which shows mitochondrial dysfunction, in denervated muscle tissues was significantly higher in myonectin-KO mice than in WT mice (Fig. 4b). Consistently, citrate synthase activity, which is a marker of intact mitochondrial function, in isolated mitochondrial fraction from gastrocnemius muscle was lower in myonectin-KO mice than in WT mice (Fig. 4c). In addition, the expression of mitochondrial complex I and complex II in isolated mitochondrial fraction from gastrocnemius muscle was lower in myonectin-KO mice than in WT mice (Fig. 4d). Thus, myonectin could exert protective function against mitochondrial dysfunction induced by muscle atrophy.

### Myonectin ameliorates DEX-induced myotube atrophy through AMPK/PGC1α pathway

To dissect the precise mechanism by which myonectin prevents muscle atrophy, C2C12 myotubes were used for analysis. We evaluated the effect of myonectin on DEX-induced myotube atrophy. Treatment with myonectin protein restored DEX-induced reduction of myotube diameter (Fig. 5a). Treatment of C2C12 myotubes with myonectin increased the expression levels of PGC1α, PGC1α4 and IGF1 in the presence or absence of DEX (Fig. 5b). Furthermore, treatment of C2C12 myotubes with myonectin protein time-dependently increased the expression levels of PGC1α and the phosphorylation levels of AMPK and acetyl CoA carboxylase (ACC) (Fig. 5c). To clarify whether the effects of myonectin were mediated by AMPK/PGC1α dependent pathway, C2C12 myotubes were transduced with adenoviral vectors expressing dominant-negative mutant of AMPK (Ad-dnAMPK) or control β-galactosidase (Ad-β-gal). Transduction of C2C12 myotubes with Ad-dnAMPK suppressed myonectin-stimulated phosphorylation of ACC, which is a downstream signal of AMPK (Fig. 5d). Transduction with Ad-dnAMPK diminished myonectin-enhanced expression of PGC1α in C2C12 myotubes (Fig. 5d). Of note, Ad-dnAMPK treatment reversed myonectin-induced increase in myotube diameter in the presence of DEX (Fig. 5e).

AMPKα subunit has two isoforms, α1 and α2[15]. To investigate the isoform-specific roles of AMPKα in regulation of PGC1α and PGC1α4 expression, C2C12 myotubes were transfected with siRNA targeting AMPKα1 or AMPKα2, or unrelated control siRNA. Treatment of C2C12 myotubes with siRNA targeting AMPKα1 and AMPKα2 reduced the expression of AMPKα1 and AMPKα2 by 76.1 ± 1.3 % and 87.8 ± 1.0 % compared with control siRNA treatment, respectively (Supplementary Fig. 9a). Knockdown of AMPKα1 did not affect myonectin-stimulated increases in PGC1α and PGC1α4 expression in C2C12 myotubes (Supplementary Fig. 9b). Importantly, ablation of AMPKα2 abolished myonectin-stimulated increases in PGC1α and PGC1α4 expression in C2C12 myotubes (Supplementary Fig. 9c). Furthermore, C2C12

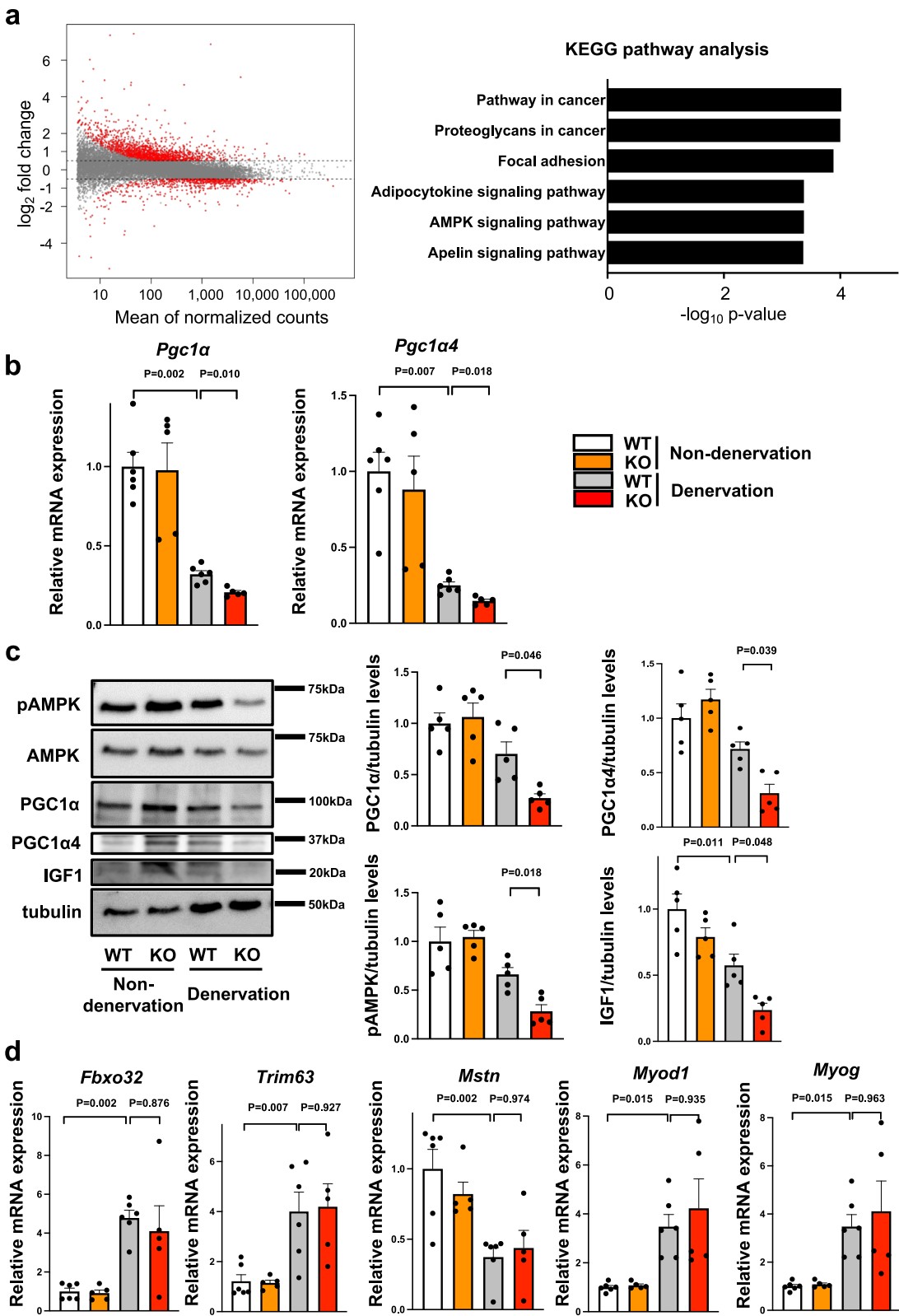

myotubes were transfected with siRNA targeting PGC1α or unrelated control siRNA. Treatment of C2C12 myotubes with siRNA targeting PGC1α reduced the expression of PGC1α and PGC1α4 by 75.5 ± 2.4% and 75.8 ± 4.9% compared with control siRNA treatment, respectively (Supplementary Fig. 10a). Knockdown of PGC1α canceled myonectin-mediated increase in myotube diameter in the presence of DEX (Fig. 5f). Moreover, ablation of PGC1α reversed myonectin-stimulated

increase in IGF1 expression in C2C12 myotubes in the presence of DEX (Supplementary Fig. 10b).

To examine whether overexpression of PGC1α4 could rescue AMPK inactivation-induced reduction of anti-atrophy effects of myonectin, C2C12 myotubes were transduced with adenoviral vectors expressing PGC1α4 (Ad-PGC1α4) or control Ad-βgal in the presence or absence of Ad-dnAMPK. Treatment of C2C12 myotubes with Ad-

**Fig. 3 | Myonectin deficiency reduces AMPK/PGC1α signaling cascade in skeletal muscle after denervation. a–d** WT and myonectin-KO mice at the age of 8–10 weeks were subjected to sciatic denervation-induced muscle atrophy operation. At 5 days after sciatic nerve denervation, the denervated and non-denervated (contralateral) gastrocnemius muscles were used for analyses. **a** Gene expression changes in denervated gastrocnemius muscle between WT and myonectin-KO mice are evaluated by RNA-seq. Left panel shows MA-plots of differentially regulated genes between myonectin-KO and WT mice. The x-axis represents the mean normalized counts and the y-axis shows the $\log_2$ fold change. Red plots show significantly regulated genes with FDR-adjusted *p* value < 0.05 and an absolute log2 fold change > 0.5. Right panel shows KEGG pathway enrichment analysis for significantly differentially regulated genes. *N* = 4 in each group. **b** The mRNA levels of *Pgc1α* and *Pgc1α4* are evaluated by quantitative PCR analysis. *N* = 6 in non-denervated and denervated WT mice. *N* = 5 in non-denervated and denervated KO mice. **c** Left panels show representative Western blot analyses of pAMPK, AMPK, PGC1α, PGC1α4, IGF1 and tubulin. Right panels show quantitative analyses of PGC1α, PGC1α4, pAMPK and IGF1 signals normalized to tubulin signal. *N* = 5 in each group. **d** The mRNA levels of ubiquitin ligase and myogenic genes. *N* = 6 in non-denervated and denervated WT mice. *N* = 5 in non-denervated and denervated KO mice. Data are presented as means ± SEM. One-way ANOVA with a post-hoc analysis for **b**–**d** was performed.

PGC1α4 increased the protein levels of PGC1α4 (Supplementary Fig. 11a). Treatment with Ad-PGC1α4 reversed the suppressive effects of AMPK inactivation on myonectin-induced increase in myotube diameter in the presence of DEX (Supplementary Fig. 11b). These data indicate that myonectin could protect against DEX-induced myotube atrophy through the AMPKα2/PGC1α4-dependent pathway.

## Myonectin supplementation attenuates denervation-induced muscle atrophy through the AMPK/PGC1α pathway

To assess the therapeutic effect of myonectin on muscle atrophy, recombinant myonectin protein was sustainably administrated to the gastrocnemius muscle of WT mice by using gelatin hydrogel immediately after surgical denervation. Myonectin administration to WT mice increased the gastrocnemius muscle weights normalized by body weights after denervation to the levels of non-denervated gastrocnemius muscle (Fig. 6a). Treatment of WT mice with myonectin did not affect body weight after denervation compared with vehicle treatment (Supplementary Fig. 12a). The weights of denervated gastrocnemius muscle seemed to be higher in myonectin-treated WT mice than in vehicle-treated WT mice, but this was not statistically significant. There were no significant differences in non-denervated gastrocnemius muscle weight between vehicle-treated WT and myonectin-treated WT mice.

In histological analysis, treatment of WT mice with myonectin significantly increased mean CSA of denervated gastrocnemius muscle (Fig. 6b). Treatment of WT mice with myonectin increased the frequency of larger CSA of denervated gastrocnemius muscle. Myonectin treatment had no effects on weights of gastrocnemius muscle and CSA of muscle in non-denervated WT mice. Myonectin treatment significantly increased mean CSA of type II muscle fibers in gastrocnemius muscle in denervated WT mice (Supplementary Fig. 13a, b). Myonectin treatment induced the larger size distribution of type II fiber CSA of gastrocnemius muscle in denervated WT mice (Supplementary Fig. 13b). On the contrary, myonectin treatment did not affect mean CSA of type I muscle fibers and its distribution in gastrocnemius muscle in denervated WT mice (Supplementary Figs. 14a and 13b). Myonectin treatment had no effects on the frequencies of type I fibers in gastrocnemius muscle in denervated WT mice (Supplementary Fig. 13c). Furthermore, administration of myonectin one day after denervation significantly increased the gastrocnemius muscle weights normalized by body weights in WT mice (Supplementary Fig. 14). Therefore, these data indicate that myonectin supplementation can be therapeutically effective at improving muscle atrophy.

Consistent with the data observed in loss-of-function experiments, myonectin treatment increased the protein levels of PGC1α and PGC1α4, and phosphorylation level of AMPK in gastrocnemius muscles after denervation (Fig. 6c). To evaluate whether AMPK is involved in the beneficial effects of myonectin on muscle atrophy, transgenic mice overexpressing dominant negative mutant form of AMPK under the control of skeletal muscle specific human α-skeletal actin (HSA) promoter (DN-AMPK Tg) were used for analysis. In contrast to WT mice, the stimulatory effects of myonectin on protein levels of PGC1α, PGC1α4 and IGF1 in gastrocnemius muscle after denervation were not observed in DN-AMPK Tg mice (Fig. 6d). Similarly, myonectin had little effects on gastrocnemius muscle weights per body weights after denervation in DN-AMPK Tg mice (Fig. 6e). Furthermore, Ad-dnAMPK or Ad-β-gal was intramuscularly administered to the gastrocnemius muscle of WT mice at 3 days before myonectin treatment. Although myonectin treatment increased the gastrocnemius muscle weights per body weights of Ad-β-gal-treated WT mice after denervation, it had no effects on gastrocnemius muscle weights per body weights of Ad-dnAMPK-treated WT mice after denervation (Supplementary Fig. 15). These results indicate that myonectin treatment protects against muscle atrophy through the AMPK/PGC1α-dependent pathway.

## Myonectin administration maintained progressive muscle atrophy in male SAMP8 and mdx mice

To evaluate the effect of myonectin on age-associated sarcopenia, senescence accelerated mouse prone (SAMP) 8 mice, which display a phenotype of accelerated aging, were subjected to intramuscular administration of Ad-myonectin or control Ad-β-gal at the age of 33 weeks. Myonectin protein levels in gastrocnemius muscle tissues were significantly reduced in SAMP8 mice compared with control SAMR1 mice at the age of 37 weeks (Supplementary Fig. 16a). Treatment of SAMP8 mice with Ad-myonectin increased the expression levels of myonectin in gastrocnemius muscle by a factor of 3.4 ± 0.5 compared with Ad-β-gal treatment (Supplementary Fig. 16b). At 4 weeks after administration of Ad-β-gal or Ad-myonectin, gastrocnemius muscle weights per body weights were significantly higher in Ad-myonectin-treated SAMP8 mice than in Ad-β-gal-treated SAMP8 mice (Fig. 7a). Body weight did not differ between Ad-β-gal-treated and Ad-myonectin-treated SAMP8 mice (Supplementary Fig. 12b). The gastrocnemius muscle weight of Ad-myonectin-treated SAMP8 mice seemed to be increased compared with that of Ad-β-gal-treated SAMP8 mice, but this was not statistically significant.

Histological analysis also exhibited that Ad-myonectin treatment increased mean CSA of gastrocnemius muscle in SAMP8 mice compared with Ad-β-gal treatment (Fig. 7b). Ad-myonectin treatment also enhanced the protein levels of PGC1α and PGC1α4 in gastrocnemius muscle in SAMP8 mice (Fig. 7c).

Finally, we assessed the effect of myonectin on muscle atrophy in mdx mice, which is a genetic muscular dystrophy model. Myonectin protein levels In gastrocnemius muscle were significantly reduced in mdx mice compared with control WT mice at the age of 8 weeks (Supplementary Fig. 17a). Mdx mice at the age of 4 weeks were subjected to intramuscular administration of Ad-β-gal or Ad-myonectin. Treatment of mdx mice with Ad-myonectin increased the expression levels of myonectin in gastrocnemius muscle by a factor of 3.7 ± 0.4 compared with Ad-β-gal treatment (Supplementary Fig. 17b). At 4 weeks after administration of Ad-β-gal or Ad-myonectin, gastrocnemius muscle weights per body weights were significantly higher in Ad-myonectin-treated mdx mice than in Ad-β-gal-treated mdx mice (Fig. 7d). Body weight did not differ between Ad-β-gal-treated and Ad-myonectin-treated mdx mice (Supplementary Fig. 12c). The gastrocnemius muscle weight tended to be higher in Ad-myonectin-treated

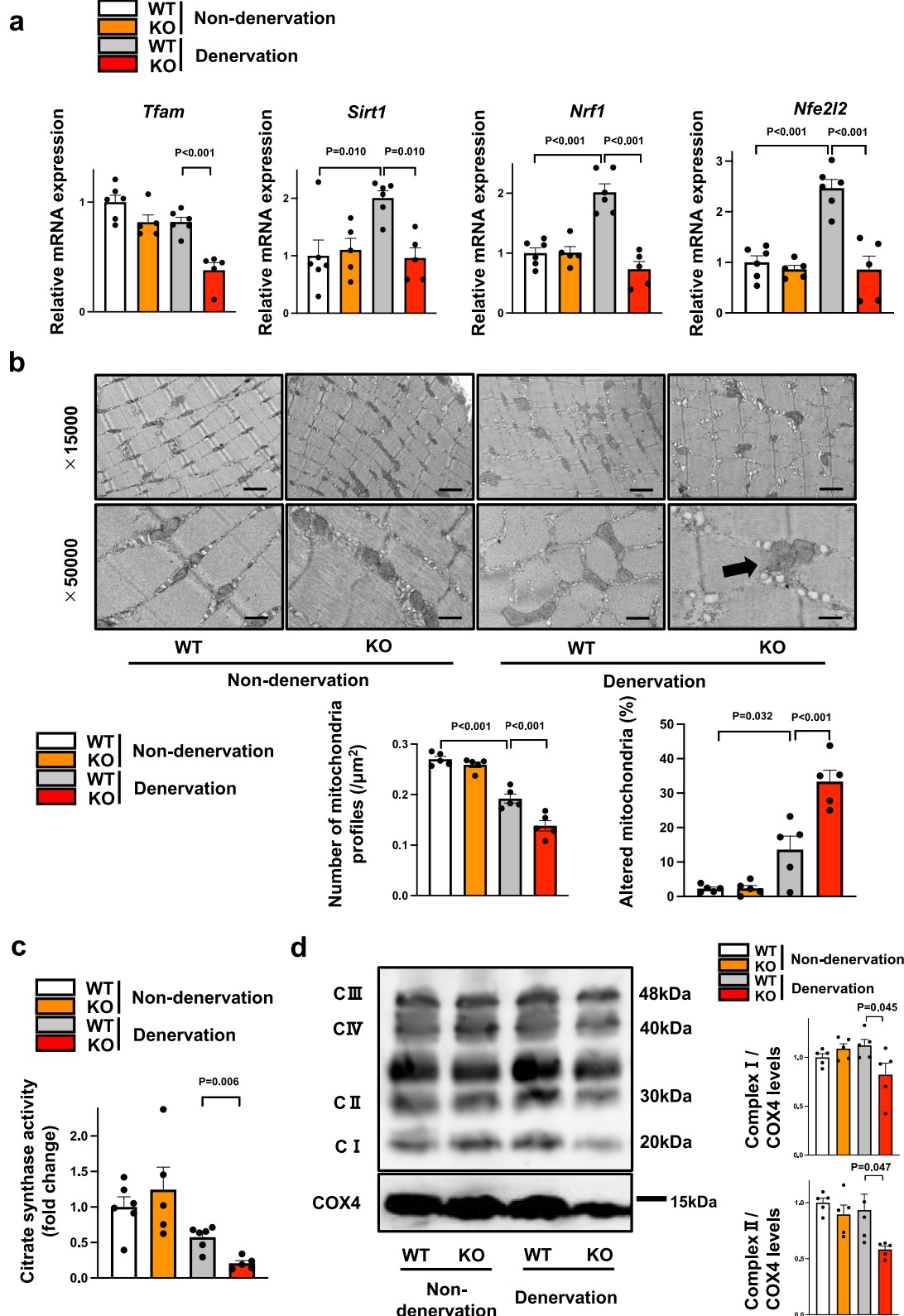

mdx mice than in Ad-β-gal-treated mdx mice, but this was not statistically significant.

Histological analysis demonstrated that Ad-myonectin treatment increased CSA of gastrocnemius muscle in mdx mice compared with Ad-β-gal treatment (Fig. 7E). The protein levels of PGC1α and PGC1α4 in gastrocnemius muscle of mdx mice were also increased by Ad-myonectin treatment (Fig. 7f). In addition, treatment of mdx mice with

Ad-myonectin treatment led to increased expression of embryonic myosin heavy chain (*Myh3*) and frequency of central nuclei-positive cells in gastrocnemius muscle tissues, which indicate the promotion of skeletal muscle regeneration (Supplementary Fig. 17c, d). Collectively, these findings indicated that myonectin treatment could be effective in ameliorating various types of muscle atrophy including age-associated sarcopenia and genetic muscle dystrophy.

**Fig. 4 | Myonectin deficiency leads to exacerbated mitochondrial dysfunction in denervated muscle. a–d** WT and myonectin-KO mice at the age of 8–10 weeks were subjected to sciatic denervation-induced muscle atrophy operation. At 5 days after sciatic nerve denervation, the denervated and non-denervated gastrocnemius muscles were used for analyses. **a** The mRNA levels of genes related to mitochondrial biogenesis were evaluated by quantitative real time PCR method. $N = 6$ in non-denervated and denervated WT mice. $N = 5$ in non-denervated and denervated KO mice. **b** Upper panels show representative electron micrographs of gastrocnemius muscles. Arrow shows altered mitochondria. Altered mitochondria is defined as a mitochondria which has partial or complete separation of the outer and inner membranes and swelling. Lower panels show quantification of intramyofibrillar mitochondria number and altered mitochondria percentage. $N = 5$ in each group. Scale bars show 1μm in X15,000 and 400 nm in X50,000. **c** Relative citrate synthase activity in isolated mitochondrial fraction from denervated gastrocnemius muscles was measured. $N = 6$ in non-denervated and denervated WT mice. $N = 5$ in non-denervated and denervated KO mice. **d** Left panel shows representative Western blot analyses of oxidative phosphorylation complexes. Right panels show quantitative analyses of the complex I and complex II signals. $N = 5$ in each group. Data are presented as means ± SEM. One-way ANOVA (**a**, **b**) with a post-hoc analysis and two-tailed unpaired Student's $t$-test (**c**, **d**) were performed.

## Discussion

In the present study, we provided the evidence that myonectin functions as a modulator of skeletal muscle mass and function. Aged mice showed marked reduction of myonectin expression in skeletal muscle. Myonectin-deficiency led to exacerbated atrophy of skeletal muscle, in particular type II muscle fiber, in aged mice, which was accompanied by reduction of muscle strength and exercise capacity. Myonectin-deficiency also resulted in aggravated skeletal muscle atrophy in sciatic denervation-induced or DEX-induced muscle atrophy models. Conversely, myonectin administration one day or immediately after denervation improved muscle atrophy in mice. In vitro experiments also demonstrated that myonectin reversed DEX-induced myofiber atrophy. Furthermore, intramuscular administration of myonectin ameliorated spontaneous progression of muscle atrophy in SAMP8 mouse models of accelerated aging or mdx mouse models of muscular dystrophy. Therefore, multiple lines of evidence indicate that myonectin can act as a protective factor against skeletal muscle dysfunction, suggesting that myonectin can represent a therapeutic target for muscle atrophy.

Mitochondrial dysfunction has been involved in skeletal muscle disorders such as sarcopenia[16]. Exercise training promotes skeletal muscle mitochondrial function, and benefits the prevention or treatment of various chronic conditions including metabolic dysfunction and sarcopenia[17]. PGC1α is a transcriptional coregulator induced by exercise training, and controls mitochondrial energy metabolism[18,19]. PGC1α transgenic mice exhibit activated mitochondrial biogenesis in skeletal muscle and enhanced fatigue-resistance[20]. PGC1α deficiency leads to exacerbated abnormalities of mitochondrial structure and function, and decreased exercise performance[21,22]. Our findings demonstrated that myonectin-deficiency contributes to reduction of PGC1α-dependent pathway and exacerbation of mitochondrial dysfunction and myofiber atrophy in denervated skeletal muscle in mice. Myonectin-deficiency also participates in enhancement of skeletal muscle atrophy and reduction of muscle strength and exercise capacity in aged mice. Thus, these findings suggest that myonectin protects against muscle dysfunction, at least in part, through modulation of PGC1α-dependent mitochondrial function.

PGC1α in skeletal muscle is activated by AMPK, cyclic AMP (cAMP)/cAMP response element binding protein (CREB) and calmodulin-dependent kinase pathways following exercise training[23–25]. Treatment of mice with AICAR, which is an activator of AMPK, leads to enhanced running endurance, which is accompanied by increased expression levels of PGC1α in skeletal muscle[26]. In agreement with these findings, our present findings indicate that AMPK functions as an upstream of PGC1α in skeletal muscle following myonectin treatment. Thus, it is conceivable that myonectin can improves skeletal muscle dysfunction through the AMPK/PGC1α-dependent mechanism. At this time nothing is known about the mechanism of how myonectin activates AMPK signaling pathway, and this needs further investigation.

PGC1α4 is reported to induce skeletal muscle hypertrophy partly via regulation of IGF1 expression[14]. Our present data documented that myonectin positively regulates protein expression of PGC1α4 and IGF1 in skeletal muscle and C2C12 myotubes. AMPKα2 is shown to regulate PGC1α4 expression in C2C12 cells[15]. Consistently, our gene knockdown experiments showed that myonectin increases PGC1α4 expression in C2C12 myotubes through AMPKα2-dependent pathway. Moreover, myonectin enhances expression of PGC1α4 and IGF1 in skeletal muscle in vivo through AMPK-dependent pathway. Thus, it is likely that myonectin increases PGC1α4 protein expression in skeletal muscle via activation of AMPKα2. In addition, siRNA-mediated knockdown of PGC1α, which also reduces PGC1α4 expression, blocks anti-atrophy effect of myonectin in C2C12 myotubes. Furthermore, overexpression of PGC1α4 rescues AMPK inactivation-induced reduction of myotube anti-atrophy action of myonectin. Taken together, these findings suggest that myonectin could prevent muscle atrophy, at least in part, through the AMPKα2/PGC1α4/IGF-1-dependent mechanisms.

Endurance exercise training is effective in reducing the risk for cardiometabolic diseases[27,28]. It has been reported that myonectin negatively regulates adipogenesis and adipose tissue accumulation[6,8]. Recently we have shown that myonectin improves myocardial ischemia-reperfusion injury in mice[9]. Of note, we have also shown that endurance exercise increases skeletal muscle and circulating levels of myonectin, and that myonectin mediates the beneficial effects of endurance exercise on the ischemic hearts. Our present data indicate that myonectin positively regulates AMPK/PGC1α signals in skeletal muscle, which can be activated by endurance exercise training. Taken together, these results suggest that myonectin treatment could mimic the salutary actions of endurance exercise training on a number of pathological conditions.

Disuse-induced muscle atrophy can occur throughout the human lifespan and decrease the quality of life[29]. PGC1α expression levels remarkably decline in response to denervation or disuse of skeletal muscle, and maintenance of PGC1α levels leads to protection against muscle atrophy. Our in vivo findings indicate that myonectin can ameliorate denervation-induced muscle atrophy through its ability to increase PGC1α expression. Glucocorticoid-induced muscle atrophy is also involved in poor quality of life and is one of serious social problems. Steroid drugs including DEX induce skeletal muscle atrophy through glucocorticoid receptor-mediated activation of FoxO, atrogin-1 and MuRF-1[30,31]. In addition, DEX attenuates PGC1α expression in L6 myotubes by reducing CREB-regulated transcription coactivator 2, which is a positive regulator of PGC1α[32]. Our gene knockdown experiments demonstrated that the suppressive effects of myonectin on myofiber atrophy are partly dependent on its ability to induce PGC1α expression. Thus, these findings suggest that myonectin could improve muscle atrophy induced by disuse and steroid use through upregulation of PGC1α expression.

Duchenne muscle dystrophy (DMD) is an early progressive and life-threatening disease. In early childhood, the patients with DMD suffer from the symptom of muscle weakening, finally leading to loss of ambulation from the ages of 8 to 12. Recently, exon-skipping therapies are developed and are effective for mitigating the symptoms of DMD patients with the mutation of dystrophin applicable to exon-skipping methods[33]. On the other hand, most of the DMD patients with the dystrophin mutation are unapplicable to exon-skipping methods[34]. In the present study, we found that myonectin treatment attenuates the progression of muscle dystrophy of mdx mice, which is a mouse

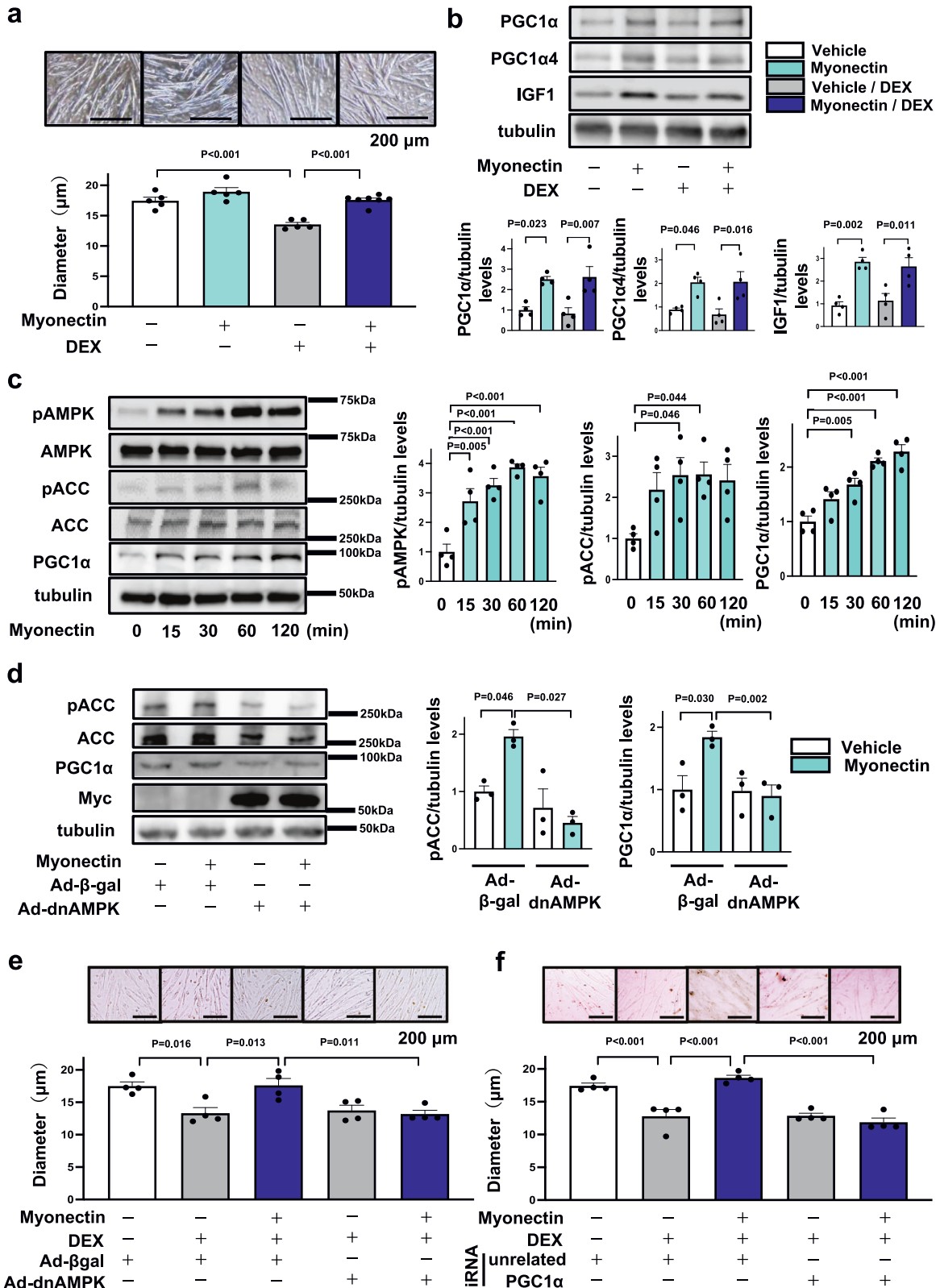

model of DMD. It has been reported that skeletal muscle specific PGC1α transgenic mice crossed with mdx mice exhibit improved muscle atrophy, running performance and plasma creatine kinase levels compared with mdx mice[35]. In addition, transient PGC1α over-expression in skeletal muscle by plasmid transfection exerts salutary actions on mitochondrial function in mdx mice[36]. Therefore, these findings indicated that myonectin could prevent muscle dystrophy of

mdx mice through PGC1α-stimulated mitochondrial biogenesis, suggesting myonectin may represent a potential therapeutic target for DMD.

In conclusion, our present study is the report that myonectin serves as a protective factor for skeletal muscle dysfunction caused by various pathophysiology conditions, such as age-associated, disuse-induced or steroid-induced muscle atrophy. Because myonectin

**Fig. 5 | Myonectin ameliorates DEX-induced atrophy of myotubes through the AMPK/PGC1α pathway. a, b** C2C12 myotubes were pretreated with myonectin protein (5 μg/ml) or vehicle for 1 hour, followed by stimulation with DEX (100 μM) or vehicle for 24 hours. **a** Upper panels show representative images of myotubes. Scale bars show 200 μm. Lower panel shows quantification of myotube diameter. *N* = 5 in each group. **b** Upper panels show representative Western blot analyses of PGC1α, PGC1α4, IGF1 and tubulin. Lower panels show quantification of PGC1α, PGC1α4 and IGF1 signals normalized to tubulin signal. *N* = 4 in each group. **c** C2C12 myotubes were treated with myonectin protein (5 μg/ml) or vehicle for the indicated lengths of time. Left panels show representative Western blot analyses of pAMPK, AMPK, pACC, ACC, PGC1α and tubulin. Right graphs show quantitative analyses of the pAMPK/tubulin, pACC/tubulin and PGC1α/tubulin signal ratios. *N* = 4 in each group. **d, e** C2C12 myotubes were pretreated with adenoviral vectors expressing dominant-negative mutant form of AMPKα2 tagged by Myc (Ad-dnAMPK) or control Ad-β-gal 24 hours prior to myonectin treatment. **d** C2C12

myotubes were treated with myonectin protein (5 μg/ml) or vehicle for 1 hour. Left panels show representative Western blots of pACC, ACC, PGC1α, Myc-Tag and tubulin. Right panels show quantitative analyses of pACC/tubulin and PGC1α/tubulin signal ratios. *N* = 3 in each group. **e** C2C12 myotubes were pretreated with myonectin protein (5 μg/ml) or vehicle for 1 hour, followed by stimulation with DEX (100 μM) or vehicle for 24 hours. Upper panels show representative images of myotubes. Scale bars show 200 μm. Lower panel shows quantification of myotubes diameter. *N* = 4 in each group. **f** C2C12 myotubes were transduced with siRNA targeting PGC1α or unrelated control siRNA at 24 hours before myonectin treatment. Upper panels show representative images of myotubes. Lower panel shows quantification of myotubes diameter. *N* = 4 in each group. Scale bars show 200 μm. Data are presented as means ± SEM. One-way ANOVA with a post-hoc analysis for **a–f** was performed. DEX, dexamethasone. Ad-dnAMPK Adenovirus vectors containing the gene for dominant-negative AMPKα2, Ad-β-gal Adenovirus vectors containing the gene for β-galactosidase.

expression in skeletal muscle was markedly reduced in association with various muscle wasting conditions, the approaches to increased myonectin production in dysfunctional muscle could be potentially valuable for prevention or treatment of age-related skeletal muscle atrophy.

## Methods

### Materials

Antibodies against phosphorylated AMPK (Thr172)(Cat. 2531 S)(diluted at 1:1000), AMPK (Cat. 2532 S)(diluted at 1:1000), phosphorylated ACC (Ser79)(Cat. 3661)(diluted at 1:1000), ACC (Cat. 3662)(diluted at 1:1000), COX4 (Cat. 4844)(diluted at 1:1000) and α-tubulin (Cat. 2144 S)(diluted at 1:1000) were purchased from Cell Signaling Technology. Antibodies against AMPKα1 (Cat. ab3759)(diluted at 1:1000), AMPKα2 (Cat. ab3760)(diluted at 1:1000) and total OXPHOS (Cat. ab110413)(diluted at 1:500) were purchased from Abcam. Antibody against myonectin was purchased from Santa Cruz Biotechnology (Cat. sc-246565)(diluted at 1:2500). Antibody against IGF1 (Cat. AF-791) was purchased from R&D Systems, Inc.(diluted at 1:500). Antibody against PGC1α (PGC1α and PGC1α4) was purchased from Calbiochem (Cat.ST1202-1SETCN). Mouse monoclonal anti-slow myosin (Clone NOQ7.5.4D) (Cat. SAB4200670)(diluted at 1:10000) and mouse monoclonal anti-fast myosin (Clone MY-32)(Cat. M4276)(diluted at 1:800) were purchased from Sigma. Goat anti-rabbit IgG HRP-linked antibody (Cat. 7074) and horse anti-mouse IgG HRP-linked antibody (Cat. 7076) were purchased from Cell Signaling Technology (diluted at 1:5000). Bovine anti-goat IgG HRP-linked antibody (Cat. sc-2384) was purchased from Santa Cruz (diluted at 1:2000). Biodegradable gelatin hydrogel was purchased form Nitta Gelatin Inc.. Dexamethasone (DEX) was purchased from Cayman Chemical Co.. Recombinant mouse myonectin protein was prepared by use of mammalian expression vector encoding full-length mouse myonectin (*Fam132b*) cDNA[9].

### Preparation of adenoviral vectors

Adenoviral vectors expressing mouse full-length mouse myonectin (Ad-myonectin), mouse PGC1α4 (Ad-PGC1α4) or β-galactosidase (Ad-β-gal) were constructed under the control of the CMV promoter. Adenoviral vectors expressing dominant negative mutant form of AMPKα2 (Ad-dnAMPK) were prepared by use of the rat AMPKα2 cDNA, whose lysine 45 residue was changed to arginine, with the c-Myc epitope tag[37].

### Animal experiments

Male C57BL/6 J wild-type (WT) mice were purchased from SLC Co Ltd.. Myonectin (*Fam132b*)-knockout (myonectin-KO) mice were originally generated by Lexicon Pharmaceuticals, Inc. (Woodlands, TX) and were obtained from Taconic Biosciences, Inc (NY, USA)[9]. Myonectin-KO mice were generated in a background of C57BL/6 and male

homozygous myonectin-KO were used in this study. Human skeletal a-actin promoter (HSA) promoter-driven dominant negative mutant form of AMPK transgenic (DN-AMPK Tg) male mice in a background of C57BL/6 were purchased from JCRB (Japanese Collection of Research Bioresources Cell Bank) Laboratory Animal Resource Bank at NIBIOHN (National Institute of Biomedical Innovation, Health and Nutrition. Osaka, Japan)[38]. Male mdx mice in a background of C57BL/6 were purchased from Chubu Kagaku Shizai Co.,Ltd.(Nagoya, Japan). Male SAMP8 mice (SAMP8/TaSlc) and male control SAMR1/TaSlc mice in a background of AKR/J were purchased from Japan SLC, Inc.(Hamamatsu, Japan). Mice were housed at 20–22 °C and 50% relative humidity in a 12 h light/dark cycle. Mice had free access to water and standard chow (CE-2, CLEA Japan Inc.). Mice were sacrificed at indicated time points after anesthesia with medetomidine, midazolam, and butorphanol at doses of 0.15, 2.0 and 2.5 mg/kg, respectively. Muscles were harvested for histological and molecular analysis. All procedures were approved by the Institutional Animal Care and Use Committee of Nagoya University School of Medicine. Mice were closely observed and euthanized quickly, at a humane end point (no locomotion or body weight loss exceeding 20% of the initial body weight).

### Age-associated muscle atrophy model

Male WT and myonectin-KO mice at the age of 80 weeks were used for analysis. Grip strength was measured to evaluate muscle quality by using a small-animal grip strength meter (Columbus Co., Largo, FL). Voluntary activity of mice was measured by wheel running system (Melquest Ltd., Toyama).

### Steroid-induced muscle atrophy model

Male WT and myonectin-KO mice at the age of 8–10 weeks were continuously treated with dexamethasone (DEX) by setting Alzet mini-osmotic minipumps (Model 2002, Durect Corp.) at the dose of 23 μg/day for 14 days.

### Sciatic denervation-induced muscle atrophy model

Denervation was performed by surgical removal of sciatic nerve from right hindlimb of male WT, myonectin-KO and DN-AMPK Tg mice at the age of 8–10 weeks[39]. In some experiments, gelatin hydrogel impregnated with recombinant myonectin (12 μg) was injected into the fascia of the gastrocnemius muscle just before or one day after denervation. In some experiments, Ad-dnAMPK or Ad-β-gal (3 × 10⁸ plaque-forming units/mouse) was injected into gastrocnemius of WT mice at 3 days before denervation. At 5 days after removal of sciatic nerve, mice were sacrificed for analysis.

### Duchenne muscle dystrophy model

Adenoviral vectors expressing myonectin (Ad-myonectin) or β-galactosidase (Ad-β-gal) (1 × 10⁸ plaque-forming units/mouse) were

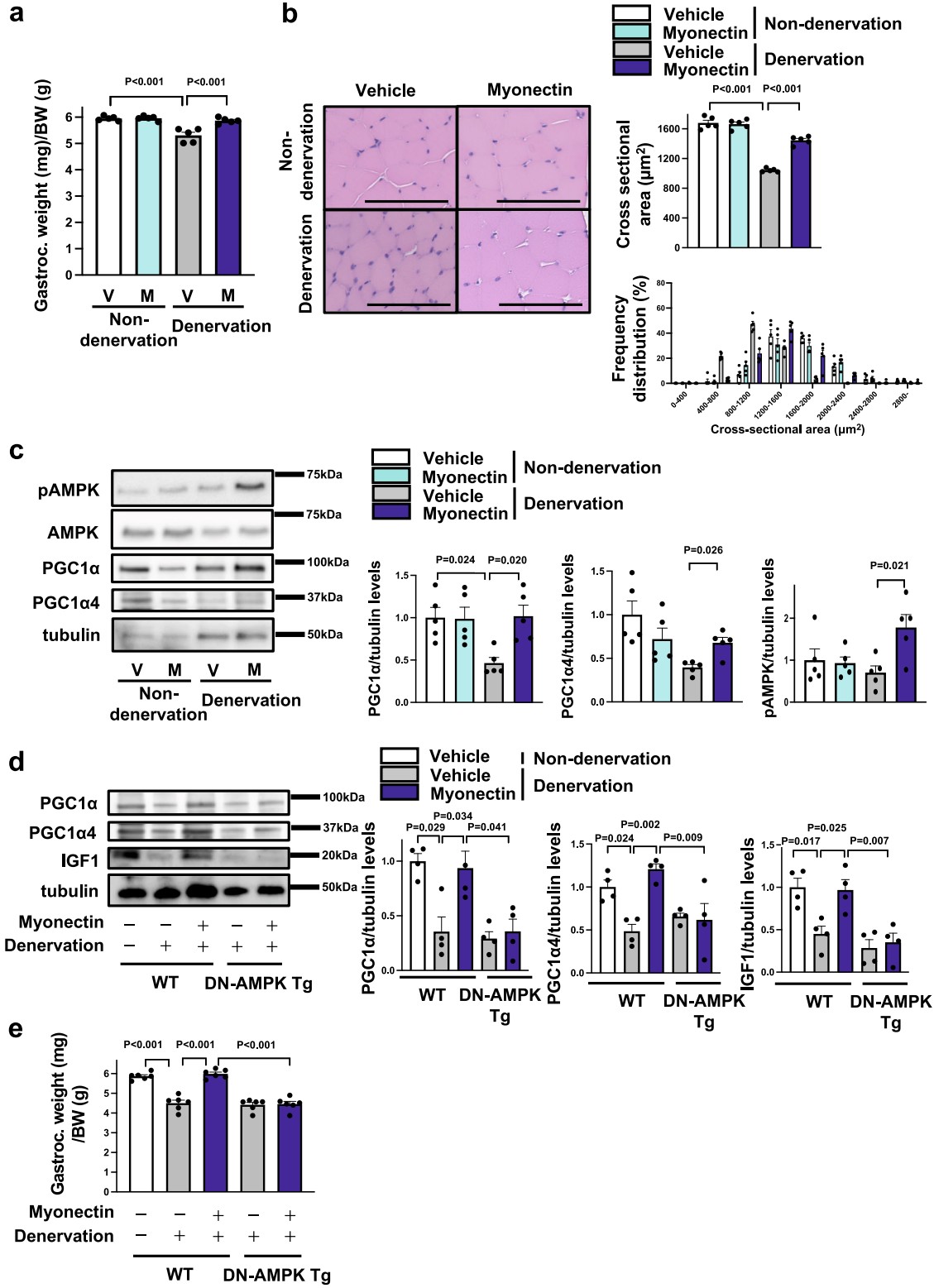

injected into gastrocnemius muscles of male C57BL/10-mdx mice at the age of 4 weeks. At 4 weeks after Ad-myonectin or Ad-β-gal treatment, mdx mice were sacrificed for analysis.

### Senescence accelerated mice (SAMP) model

Ad-myonectin or Ad-β-gal ($1 \times 10^8$ plaque-forming units/mouse) was injected into gastrocnemius muscles of male SAMP8 and control SAMR1 mice at the age of 33 weeks. At 4 weeks after treatment with Ad-myonectin or Ad-β-gal, SAMP8 mice were sacrificed for analysis.

### Grip strength

Mice were allowed to grasp a horizontal grid connected to a small-animal grip strength meter (Columbus Co., Largo, FL) with forelimbs or all four limbs[40]. The force applied to the grid each time before the

**Fig. 6 | Myonectin supplementation restores denervation-induced muscle atrophy through the AMPK/PGC1α pathway. a–c** WT mice at the age of 8–10 weeks were subjected to sciatic denervation-induced muscle atrophy operation. Gelatin hydrogel impregnated with recombinant myonectin (12 µg) or vehicle was injected into the fascia of the denervated gastrocnemius muscle just before denervation. At 5 days after sciatic nerve denervation, the denervated and non-denervated gastrocnemius muscles of myonectin- or vehicle-treated WT mice were used for analysis. **a** Gastrocnemius muscle weight/body weight ratio of myonectin- or vehicle-treated WT mice. *N* = 5 in each group. **b** Left panels show representative cross sectional muscle fiber images of myonectin- or vehicle-treated WT mice. Scale bars show 100 µm. Right panels show quantitative analyses of mean cross sectional area (CSA) and CSA distribution of gastrocnemius muscle in myonectin- or vehicle-treated WT mice. *N* = 5 in each group. **c** Left panels show representative Western blot analyses of pAMPK, AMPK, PGC1α, PGC1α4 and tubulin. Right panels show quantitative analyses of the PGC1α, PGC1α4 and pAMPK signals normalized to tubulin signal. *N* = 5 in each group. **d, e** WT and DN-AMPK Tg mice at 8–10 weeks were subjected to sciatic denervation-induced atrophy operation. Gelatin hydrogel impregnated with recombinant myonectin (12 µg) or vehicle was injected into the fascia of the denervated gastrocnemius muscle just before denervation. At 5 days after sciatic nerve denervation, WT and DN-AMPK Tg mice were sacrificed, and denervated and non-denervated gastrocnemius muscles were used for analysis. **d** Left panels show representative Western blot analyses of PGC1α, PGC1α4, IGF1 and tubulin. Right panel shows quantitative analyses of the PGC1α, PGC1α4 and IGF1 signals normalized to tubulin signal. *N* = 6 in each group. **e** The ratio of gastrocnemius muscle to body weight in myonectin- or vehicle-treated WT and DN-AMPK Tg mice. *N* = 6 in each group. Data are presented as means ± SEM. Two-tailed unpaired Student's *t*-test (**c**) and one-way ANOVA with a post-hoc analysis (**a, b, d, e**) were performed. Gastroc. gastrocnemius muscle, BW body weight, V vehicle, M myonectin, DN-AMPK Tg transgenic mice overexpressing dominant negative mutant form of AMPK by a control of skeletal muscle specific human α-skeletal actin (HSA) promoter.

animal lost its grip was recoded in Newton. The average of the five measurements was normalized to the whole-body weight.

### Voluntary wheel running activity

Mice were individually housed in a cage with a running wheel (Melquest Ltd., Toyama, Japan). Each wheel had a magnetic indicator and a hall effect sensor that connected to a computer interface and recorded wheel revolutions. Voluntary wheel running data (number of rotations) were collected daily, and mice were checked to ensure that the wheel was still functioning properly. A five-day introductory period was provided prior to data measurements.

### Histology and cross-sectional area (CSA) analysis

Excised gastrocnemius and soleus tissues were fixed with 4% paraformaldehyde and embedded in paraffin. Tissue samples were cut at a thickness of 6 µm and the sections were stained with hematoxylin-eosin (HE). The mean CSA of fibers was determined using the Image J software (National Institute of Health). Immunohistochemical staining was performed with a monoclonal antibody against skeletal slow (NOQ7.5.4D, Sigma) and fast myosin (MY-32, Sigma).

### Quantitative real-time PCR (qPCR)

The RNeasy Mini kit (Qiagen) was used to extract total mRNA from gastrocnemius muscle and cells. cDNA was synthesized using a ReverTra Ace kit (TOYOBO). Quantitative PCR analysis was performed with a BioRad real-time PCR detection system (TOYOBO). The qPCR primers are listed in Supplementary Table. Expression levels of examined genes were divided by the corresponding level of 36B4, and presented relative to the control.

### RNA-Seq analysis

RNA sequencing libraries for each sample were prepared with 1 µg total RNA using the Illumina TruSeq Stranded mRNA Library Prep Kit according to the manufacturer's instructions, and 150 bp paired-end sequencing was completed on the Illumina NovaSeq 6000. Data analysis of RNA-seq was performed with some modifications[41]. Briefly, quality of raw sequence data was assessed using FastQC (Version: FastQC 0.10.0). FASTQ files were trimmed for adapters and Phred score (>20) using TrimGalore! (0.6.4). Trimmed sequenced reads were aligned to the mouse genome assembly (GRCm39 GENCODE vM30) using STAR (2.7.3a)[42]. Aligned reads were used to quantify mRNA with HTSeq-count(version 0.11.2)[43]. Differential gene expression analysis across samples was performed using DESeq2 package (1.32.0) on protein-coding genes[44]. Genes were selected as differentially expressed when the FDR-adjusted *p* value was below 0.05 and an absolute log2 fold change was above 0.5. Analysis were conducted using R (4.1.0). Kyoto Encyclopedia of Genes and Genomes (KEGG) pathway enrichment analysis was performed using the

online bioinformatic tool, Database for Annotation, Visualization and Integrated Discovery (DAVID, v6.8) for differentially expressed genes at an FPKM (Fragments Per Kilobase of Kilobase of exon per Million fragments mapped) value of ≥1 in at least three of the samples.

### Western blot analysis

Skeletal muscle and C2C12 cell samples were solubilized with lysis buffer (Cell Signaling Technology, Cat 9803 S) and protease inhibitor cocktail (Roche, Cat 11697498001). The protein concentration was determined by a BCA protein assay kit (Thermo Scientific). The equal amounts of proteins were separated by denaturing SDS-PAGE, followed by transfer onto PVDF membrane (GE Healthcare). The membranes were exposed to primary antibody followed by incubation with the HRP-conjugated secondary antibody. The protein signal was detected by ECL prime system (GE Healthcare). The expression levels were determined by measurement of the corresponding band intensities using image J software (National Institute of Health)[45]. The relative values of the corresponding band were evaluated by relative intensities to α-tubulin signal.

### Transmission electron microscopy analysis

Gastrocnemius muscles were trimmed intro approximately 1.0 mm³ strips and fixed in a mixture of 2% glutaraldehyde and 2% paraformaldehyde for 18–20 h, followed by 2% osmium tetroxide for 2 h. Then, the tissues were dehydrated, and embedded in epoxy. Ultra-thin sections (80 nm thick) were treated with lead citrate, and observed using a transmission electron microscope (Hitachi, Japan). A minimum of 10 photomicrographs were taken randomly from each sample. The number of mitochondria was counted on micrographs in a blinded fashion[46].

### Morphometric analysis of mitochondrial alternations

Mitochondria was defined as altered according to criteria being validated by previous morphological studies[47] as follows: (1) significantly decreased electron density of the matrix (dilution, vacuolization, cavitation); (2) fragmented and ballooned cristae (intracristal swelling); (3) partial or complete separation of the outer and inner membranes; (4) mitochondrial swelling. Accordingly, the following data were calculated: (1) density of mitochondria, (2) percentage of altered mitochondria, (3) mitochondrial swelling, assessed by measuring the maximum and minimum mitochondrial diameter.

### Citrate synthase activity

Citrate synthase activity was determined with MitoCheck Citrate Synthase Activity Assay Kit (Cayman Chemicals, Ann Arbor, MI) following the manufacturer's instructions. Citrate synthase activity was further normalized to protein content.

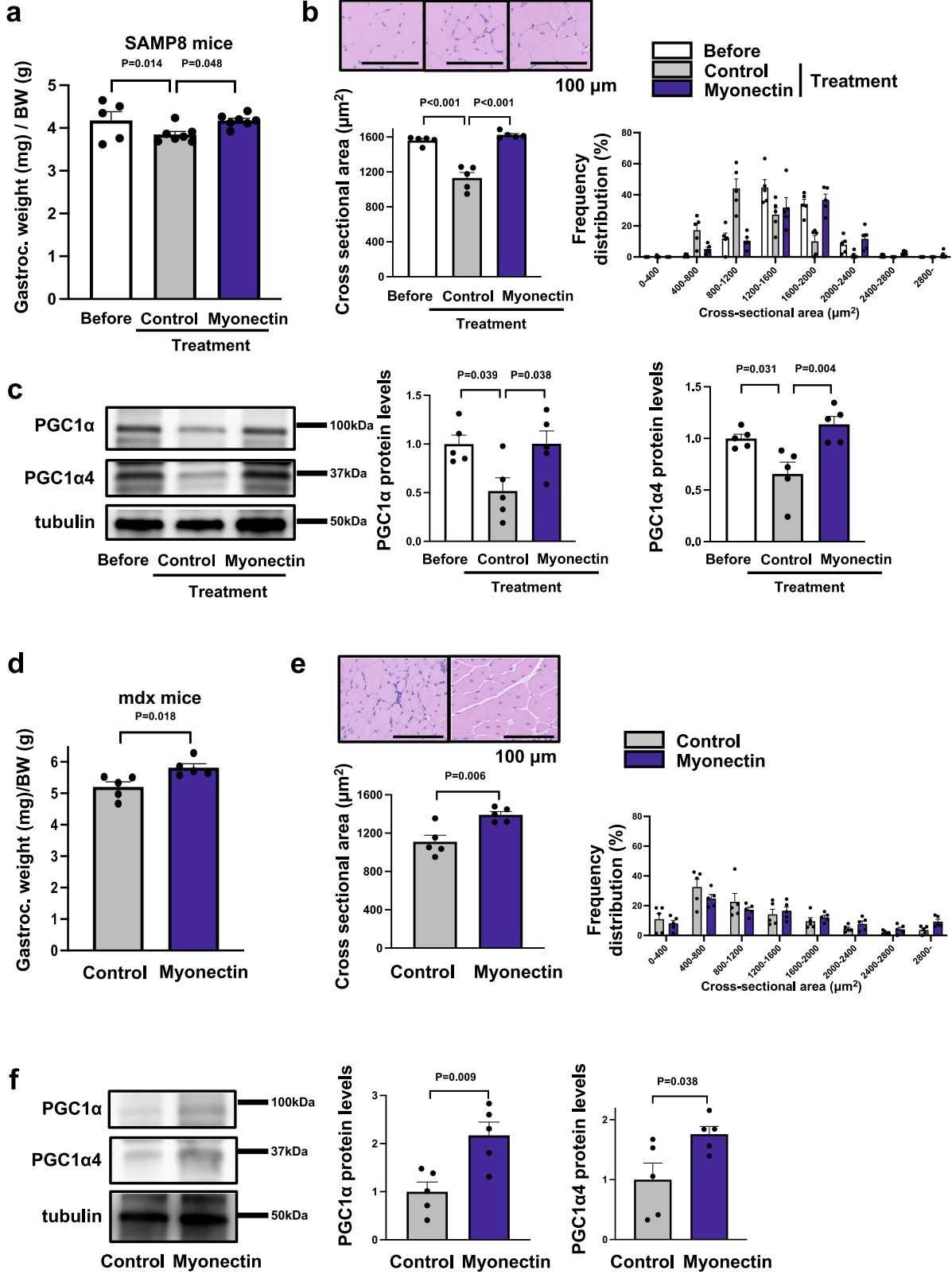

## Cell culture

C2C12 cells were purchased from American Type Culture Collection (ATCC) (CRL-1772). Myoblasts were cultured in low glucose DMEM (Sigma-Aldrich Co. USA) supplemented with 10% fetal bovine serum and 1% penicillin-streptomycin (pen/strep). Myoblasts were differentiated into myotubes by switching to differentiation medium (DMEM, 2% horse serum, 1% pen/strep) for 5 days. C2C12 myotubes were pretreated with myonectin protein (5 μg/ml) or vehicle for 1 hour, followed by stimulation with DEX (100 μM) or vehicle for 24 hours. For gene ablation experiments, these cells were transfected with small interfering RNAs (siRNAs) targeting PGC1α (L040773-01-0005), AMPKα1 (L-041035-00-0005), AMPKα2 (L-040809-00-0005) and unrelated (D-001810-10-05) siRNAs (Dharmacon Inc., Lafayette, CO) by Lipofectamine 2000 regent (Invitrogen, Life Technologies, Grand Island, NY) according to the manufacturer's protocol. At 24 hours after transfection, cells were

**Fig. 7 | Myonectin administration improves muscle atrophy in SAMP8 and Mdx mice. a**–**c** Ad-myonectin (myonectin) or Ad-β-gal (control) was intra-muscularly injected into SAMP8 mice at the age of 33 weeks. At 4 weeks after treatment with myonectin or control, SAMP8 mice were sacrificed, and the gastrocnemius muscles were used for analysis. **a** Gastrocnemius muscle weight/body weight ratio of myonectin-treated or control-treated SAMP8 mice. $N = 5$ in SAMP8 mice before treatment. $N = 7$ in SAMP8 mice after myonectin or control treatment. **b** Upper left panels show representative images of muscle fibers from SAMP8 mice before treatment or after treatment with myonectin or control. Scale bars show 100 μm. Lower panels show quantification of mean cross sectional area (CSA) and CSA distribution of SAMP8 mice before treatment or after treatment with myonectin or control. $N = 5$ in each group. **c** Left panels show representative Western blot analyses of PGC1α, PGC1α4 and tubulin in SAMP8 mice before treatment or after treatment with myonectin or control. Right graphs show quantitative analyses of PGC1α/tubulin and PGC1α4/tubulin signal ratios. $N = 5$ in each group. **d**–**f** Ad-myonectin (myonectin) or Ad-β-gal (control) was injected into gastrocnemius muscles of mdx mice at the age of 4 weeks. At 4 weeks after treatment with myonectin or control, mdx mice were sacrificed. **d** Gastrocnemius muscle weight/body weight ratio of myonectin-treated or control-treated mdx mice. $N = 5$ in each group. **e** Upper panels show representative images of muscle fibers from myonectin-treated or control-treated mdx mice. Scale bars show 100 μm. Lower panels show quantitative analysis of mean CSA and CSA distribution of myonectin-treated or control-treated mdx mice. $N = 5$ in each group. **f** Left panels show representative Western blot analyses of PGC1α, PGC1α4 and tubulin. Right graphs show quantitative analyses of the PGC1α/tubulin and PGC1α4/tubulin signal ratios of myonectin-treated or control-treated mdx mice. $N = 5$ in each group. Data are presented as means ± SEM. One-way ANOVA with a post-hoc analysis for **a**–**c**. Two-tailed unpaired Student's *t*-test for **d**–**f**. Gastroc. gastrocnemius muscle, BW body weight, SAMP8 senescence accelerated mouse prone 8, Ad-β-gal Adenoviral vectors containing the gene for β-galactosidase, Ad-myonectin Adenovirus vectors containing the gene for myonectin.

subjected to the DEX-induced atrophy experiment. The myotube diameter was determined from the averaged diameter of myotube at least in ten high power field per well using Image J software (National Institute of Health)[45].

## Statistics
Data are shown as mean ± S.E.M. The differences between two groups for variables with normal distributions were evaluated by unpaired Student's *t*-test. Differences between three or more groups were evaluated using one-way analysis of variance, with a post-hoc Tukey's test or Games-Howell test, deemed appropriate in the presence of heterogeneity of variance. A $P$ value $< 0.05$ denoted the presence of a statistically significant difference. All statistical calculations were performed by using IBM SPSS Statistics 28 software (SPSS Inc).

## Reporting summary
Further information on research design is available in the Nature Portfolio Reporting Summary linked to this article.

## Data availability
Any additional information of data can be available from the corresponding authors on request. The RNA sequencing data generated in this study have been deposited in the Gene Expression Omnibus (GEO) database under accession code GSE233328. Source data are provided with this paper.

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

## Acknowledgements

We gratefully thank for the technical assistance of Yoko Inoue. This work was supported by Grant-in-Aid for Scientific Research A (2620H005710), Grant-in-Aid for Challenging Exploratory Research (2620K21753) and grants from Takeda Science Foundation (2600007594 and 2600007051) to N Ouchi (N.O.). K.O was supported with Grant-in-Aid for Scientific Research C (2621K08101). This work was supported, in part, by JST CREST (grant number JPMJCR19H4)(T.M.).

## Author contributions

Y.O. designed the research study, conducted the experiments, acquired the data, analyzed the data and wrote the manuscript. K.O. designed the research study, conducted the experiments, acquired the data, analyzed the data, provided expertise related to the experiments and wrote the manuscript. N.O. (N. Otaka), H.K., T.T., L.F., K.T., M.T. (M. Tatsumi), S.I. conducted the experiments, acquired the data, analyzed the data. K.K. analyzed RNA-seq data. M.T. (M. Takefuji), Y.S., Y.K.B., A.I., M.K. and T.M. designed the research study, provided expertise related to the experiments. S.M. provided DN-AMPK Tg mice and provided expertise related to the experiments. N.O. (N. Ouchi) designed the research study, analyzed the data, provided expertise related to the experiments and wrote the manuscript. All authors participated in interpreting the results and revising the manuscript.

## Competing interests

The authors declare no competing interests.
