## [Peer Review File · Nature Communications]

Myonectin protects against skeletal muscle dysfunction in male mice through activation of AMPK/PGC1 α pathwayREVIEWER COMMENTS

Reviewer #1 (Remarks to the Author):

Overall a very well written paper highlighting the potential for targeting myonectin to treat muscle atrophy. The authors have provided some compelling data using both in vivo and in vitro models to show that myonectin is able to prevent muscle atrophy.

Specific Comments

The mechanism of action needs more discussion. It is unclear how the AMPK/PGC1a axis is contributing to the protection. The authors have shown that both the full length PGC1a and the alternatively spliced short isoform PGC1a4 is down regulated in the absence of myonectin. However it is the PGC1a4 that has been shown to regulate myocyte size in part via regulation of myostatin and igf-1 for example. Therefore some discussion or data should be provided to see if the short form is being also regulated in the gain of function experiments. As it is therefore unclear how the full length PGC1a is affecting cell size.

Figure 4B. The authors should indicate or describe what they are referring to in the TEM images.

Figure 5. The authors should provide mRNA and western blotting data to confirm the efficiency of knockdown at both the mRNA and protein level. This will also, help to confirm the specificity of the PGC-1a antibody, as this has been a challenge for many of the commercial antibodies.

Minor Comment

Results Section: The last section of the results is titled "Myonectin administration restores muscle atrophy in SAMP8 and mdx mice". The use of the word "restores" seems not be the accurate word choice given the results.

Reviewer #2 (Remarks to the Author):

In this manuscript, Ozaki and co-authors examine the impact of the muscle-secreted factor myonectin in different models of muscle atrophy. They report that myonectin knock-out mice are prone to atrophy induced by aging, denervation, and treatment with dexamethasone. Moreover, they find that recombinant/transgenic myonectin reduces atrophy induced by denervation and associated with senescence and muscular dystrophy. They also provide a possible mechanism for the protection action of myonectin based on the stimulation of AMPK via an unknown receptor and subsequent increase in PGC-1 α levels and function (mitochondrial biogenesis). Overall, this is an interesting manuscript that expands the knowledge on the protective action of myonectin. There are however major limitations of the current manuscript. As explained in detail here below, these include the lack of analysis of myofiber size, myofiber type, and myofiber number in the models of muscle atrophy and the low number of animals (n=4) used for many of the in vivo studies and/or subsequent analyses.

Major points:

1) Figures 1, 2, 6, and 7 report the weight of the gastrocnemius and soleus muscles normalized by body weight (BW) in different models of muscle atrophy. Figures 1 and 2 analyze the propensity of myonectin knock-out mice to undergo wasting in response to aging (Figure 1) and denervation and dexamethasone treatment (Figure 2). However, considering that myonectin is known to regulate whole-body metabolism, it remains undetermined whether there are changes in BW in response to myonectin knockout. In that case, the authors should show separate graphs reporting the BW and the raw (not-normalized) data for the weight of the soleus and gastrocnemius muscles. This applies also to Figures 6 and 7. Likewise, the muscle force measured with grip strength assays (Fig. 1D) should be reported as raw data and also as specific force (=normalized by the muscle weight or cross-sectional area).

2) Muscle weight data and muscle cross-sectional area are interesting but incomplete. The authors should provide detailed histological analyses of the gastrocnemius and soleus muscles for the different models of muscle atrophy they utilized in Figures 1, 2, 6, and 7. Specifically, the authors should analyze the size distribution (gaussian plots) of different

myofiber types, the percentage of myofiber types, and the total number of myofibers per muscle in soleus and gastrocnemius muscles from wild-type and myonectin knock-out mice in control conditions as well as when challenged by aging (Fig. 1), and by denervation and dexamethasone treatment (Fig. 2). Likewise, the analysis of myofiber size, type, and number of the gastrocnemius and soleus muscles should also be done for studies in Figures 6 and 7 that have analyzed the impact of recombinant/transgenic myonectin on muscle atrophy induced by denervation, senescence (SAMP8), and muscular dystrophy (mdx mice).

3) Many of the in vivo studies (such as in Fig. 1) have utilized a low number of animals (n=4). This number of animals is insufficient especially for muscle functional assays (Fig. 1E), which have a high intrinsic variability. In figures 2, 6, and 7 more than 4 animals have been utilized in A but only muscles from n=4 were analyzed in B. It would be good if the authors could analyze the muscles from all the animals utilized.

4) The authors suggest that myonectin prevents muscle wasting via activation of P-AMPK and PGC1a. However, they experimentally test the relevance of only AMPK downstream of myonectin. Notably, AMPK activation has been previously found to induce myofiber atrophy rather than protecting from it, and these effects have been ascribed to the capacity of AMPK to reduce mTOR activity and to induce autophagy. Therefore it remains unclear how myonectin protects from wasting via AMPK activation.

5) The PGC1a4 isoform (which induces hypertrophy) could provide a possible link between myonectin, AMPK, and protection from wasting but this link is not explored beyond the qPCR in Fig. 3B. In particular, the western blots have monitored general PGC1a levels (rather than levels of the PGC1a4 isoform) and no rescue experiment has tested the functional interaction between myonectin and PGC1a4. How myonectin-AMPK would specifically regulate PGC1a4 also remains unclear.

6) The authors report that myonectin mRNA levels decline with aging in soleus and gastrocnemius muscles (Fig 1A). Do myonectin mRNA levels change also in the other models of atrophy here utilized (denervation, dexamethasone treatment, and SAMP8 and MDX mice)? Because anti-myonectin antibodies are available, the authors could also explore

whether myonectin protein levels change consistently with mRNA levels.

7) Muscle force has been assessed only for myonectin KO mice with aging (Fig. 1) but not for the other models: could the authors include muscle functional data also for dexamethasone-induced atrophy (WT versus myonectin KO mice) as well as for SAMP8 and MDX mice?

8) The assays in Fig. 4C-D (analysis of citrate synthase activity and mitochondrial complex abundance) and Fig. 6C lack the WT and KO control from non-denervated animals. Moreover, there is no loading control for the blot in Fig. 4B.

Other points:

9) Fibrosis is an important component especially of sarcopenia and muscular dystrophy therefore it would be interesting if the authors could test whether myonectin impacts the extent of fibrosis.

10) The extent of body weight loss induced by dexamethasone treatment (Fig. 2) should be shown to determine whether myonectin knockout mice undergo similar whole-body wasting compared to controls in response to treatment with this drug.

11) The RNA-seq studies in Figure 3A should be reported in a more comprehensive manner apart the few examples of genes shown in Fig. 3B-C. It would be informative to report the gene categories regulated by myonectin.

12) The studies with SAMP8 mice (Fig. 7A-C) are based on the transgenic delivery of myonectin versus control beta-Gal in 33-week-old mice and their subsequent analyses 4 months later. As additional control, the authors have examined animals before transgenic expression (33-week-old). It seems that only control animals transfected with beta-Gal differ from the control mice before treatment. Therefore, it is unclear whether there is an effect of myonectin or rather a detrimental effect of the beta-Gal control.

Do the authors have an untreated control of the same age of the animals that have been sacrificed (37-weeks)? If so, is there a difference in muscle wasting when comparing mice

with transgenic myonectin expression versus untreated controls at the same age?

13) Fig. 7D-F: MDX mice are characterized by cycles of damage and regeneration. It is unclear how myonectin would be impacting these pathogenic processes in MDX mice. A detailed analysis of muscle regeneration would be needed if the authors want to report in a meaningful way the impact of myonectin on these models, including monitoring regeneration/myogenesis markers such as embryonic MHC and the number of centrally-nucleated myofibers.

Minor points:

14) The H&E images of muscle cross-sections are of low resolution and hence not informative: they should be substituted with high-resolution histological images.

15) The EM images in Fig. 4B are at low resolution and/or of insufficient contrast to discern ultrastructural defects in mitochondria – better images should be included.

16) The original paper on PGC1a4 should be cited rather than a commentary on that paper.

17) The word “strengthen” has been used rather than “strength” and this should be corrected.

18) There is no source file reporting the primary data. From the “source file” link, I can download only another copy of the same file that can be downloaded from the “supplementary dataset” link. Likewise, the full RNA-seq dataset reported in Fig. 3 should be made publicly available.

To Reviewer #1

“Overall a very well written paper highlighting the potential for targeting myonectin to treat muscle atrophy. The authors have provided some compelling data using both in vivo and in vitro models to show that myonectin is able to prevent muscle atrophy.”

Response: We thank the reviewer for these positive comments.

Specific Comments

“1. The mechanism of action needs more discussion. It is unclear how the AMPK/PGC1 α axis is contributing to the protection. The authors have shown that both the full length PGC1 α and the alternatively spliced short isoform PGC1 α 4 is down regulated in the absence of myonectin. However it is the PGC1 α 4 that has been shown to regulate myocyte size in part via regulation of myostatin and igf-1 for example. Therefore some discussion or data should be provided to see if the short form is being also regulated in the gain of function experiments. As it is therefore unclear how the full length PGC1 α is affecting cell size.”

Response: We thank the reviewer for these important suggestions. As the reviewer mentioned, PGC1 α 4 is reported to regulate myocyte size partly via regulation of IGF1 signal. Thus, we evaluated protein levels of PGC1 α 4 and IGF1 in denervated muscle of WT and myonectin KO mice. The protein levels of PGC1 α 4 and IGF1 in denervated skeletal muscle were significantly lower in myonectin KO mice than in WT mice (new Figure 3c). Treatment of WT mice with myonectin protein increased the protein levels of PGC1 α 4 and IGF1 in denervated skeletal muscle (new Figures 6c and 6d). Furthermore, our in vitro study demonstrated that treatment of C2C12 myotubes with myonectin increased the protein levels of PGC1 α 4 and IGF1 in the presence or absence of DEX (new Figure 5b). Therefore, these data suggest that myonectin could reduce muscle atrophy through upregulation of PGC1 α 4 and IGF1 in skeletal muscle. These findings are presented in Result section and included as new Figures 3c, 5b, 6c and 6d in the revised manuscript.

We also dissected the mechanism of how myonectin-AMPK could regulate PGC1 α 4 expression. In C2C12 myotubes, myonectin-stimulated increase in PGC1 α 4 expression was canceled by siRNA-mediated knockdown of AMPK α 2 but not knockdown of AMPK α 1 (new Supplementary Figure 9). In addition, the stimulatory effects of myonectin on PGC1 α 4 protein level in skeletal muscle after denervation were not observed in dominant negative mutant form of AMPK transgenic mice (new Figure 6d). Thus, it is likely that myonectin increases PGC1 α 4 protein level in skeletal muscle via activation of AMPK α 2. These findings are presented in Result section and included as new Supplementary Figure 9 and new Figure 6d in the revised manuscript.

Moreover, we tested the functional interaction between myonectin and PGC1 α 4. Treatment of C2C12 myotubes with siRNA targeting PGC1 α reduced the expression of both PGC1 α and PGC1 α 4 compared with control siRNA treatment (new Supplementary Figure 10a). Knockdown of PGC1 α canceled myonectin-mediated increase in myotube diameter in the presence of DEX (Figure 5f). Knockdown of PGC1 α also reversed myonectin-stimulated increase in IGF1 expression in C2C12 myotubes in the presence of DEX (new Supplementary Figure 10b). Furthermore, overexpression of PGC1 α 4 by adenoviral vector system reversed the suppressive effects of AMPK inactivation on myonectin-induced increase in myotube diameter in the presence of DEX (new Supplementary Figure 11). These data indicate that myonectin could protect against DEX-induced myotube atrophy through the

AMPK α 2/PGC1 α 4-dependent pathway. These findings are presented in Result section and included as new Supplemental Figures 10 and 11, and Figure 5f in the revised manuscript.

Taken together, our findings suggest that myonectin could prevent muscle atrophy through the AMPK α 2/PGC1 α 4/IGF-1-dependent mechanisms. These points are addressed in Discussion section in the revised manuscript.

“2. Figure 4B. The authors should indicate or describe what they are referring to in the TEM images.”

Response: We thank the reviewer for the kind suggestion. We show which mitochondria is altered mitochondria by the black arrow in new Figure 4b.

“3. Figure 5. The authors should provide mRNA and western blotting data to confirm the efficiency of knockdown at both the mRNA and protein level. This will also, help to confirm the specificity of the PGC-1 α antibody, as this has been a challenge for many of the commercial antibodies.”

Response: We thank the reviewer for the important suggestion. We evaluated knockdown efficiency of siRNA targeting PGC1 α in protein levels in C2C12 myotubes by Western blot analysis. Treatment with siRNA targeting PGC1 α reduced protein levels of PGC1 α and PGC1 α 4 by 75 % and 76 % compared with treatment with control siRNA. These results are now included as new Supplementary Figure 9a in the revised manuscript.

Minor Comment

“Results Section: The last section of the results is titled “Myonectin administration restores muscle atrophy in SAMP8 and mdx mice”. The use of the word “restores” seems not be the accurate word choice given the results.”

Response: We thank the reviewer for carefully reading our manuscript. We changed the sentence to “Myonectin administration maintained progressive muscle atrophy in SAMP8 and mdx mice” in the revised manuscript.

To the Reviewer #2

“In this manuscript, Ozaki and co-authors examine the impact of the muscle-secreted factor myonectin in different models of muscle atrophy. They report that myonectin knock-out mice are prone to atrophy induced by aging, denervation, and treatment with dexamethasone. Moreover, they find that recombinant/transgenic myonectin reduces atrophy induced by denervation and associated with senescence and muscular dystrophy. They also provide a possible mechanism for the protection action of myonectin based on the stimulation of AMPK via an unknown receptor and subsequent increase in PGC-1 α levels and function (mitochondrial biogenesis). Overall, this is an interesting manuscript that expands the knowledge on the protective action of myonectin. There are however major limitations of the current manuscript. As explained in detail here below, these include the lack of analysis of myofiber size, myofiber type, and myofiber number in the models of muscle atrophy and the low number of animals (n=4) used for many of the in vivo studies and/or subsequent analyses.”

Response: We thank the reviewer for the thoughtful comments. According to the reviewer's suggestion, we added substantial histological analyses, such as myofiber size, myofiber type and myofiber number in muscle atrophy models to the revised manuscript. In addition, we increased the number of animals in the case of the low number of animals (n=4).

Major points:

“1) Figures 1, 2, 6, and 7 report the weight of the gastrocnemius and soleus muscles normalized by body weight (BW) in different models of muscle atrophy. Figures 1 and 2 analyze the propensity of myonectin knock-out mice to undergo wasting in response to aging (Figure 1) and denervation and dexamethasone treatment (Figure 2). However, considering that myonectin is known to regulate whole-body metabolism, it remains undetermined whether there are changes in BW in response to myonectin knockout. In that case, the authors should show separate graphs reporting the BW and the raw (not-normalized) data for the weight of the soleus and gastrocnemius muscles. This applies also to Figures 6 and 7. Likewise, the muscle force measured with grip strength assays (Fig. 1D) should be reported as raw data and also as specific force (=normalized by the muscle weight or cross-sectional area).

Response: We thank the reviewer for these important suggestions. The raw data of body weight, muscle weight, grip strength and grip strength normalized by the muscle weight are now included in new Supplementary Figures 2 and 12 in the revised manuscript. Although aged myonectin-KO mice showed increased body weight compared with aged WT mice, non-normalized gastrocnemius muscle weight was significantly lower in aged myonectin-KO mice than in aged WT mice. In addition, non-normalized maximal force of grip strength in 4 limbs was significantly lower in aged myonectin-KO mice than in aged WT mice. Thus, it is likely that myonectin positively regulates skeletal muscle mass and strength in aged mice.

Regarding denervation and dexamethasone (DEX) treatment models, body weight did not differ between WT and myonectin-KO mice after denervation or DEX treatment. Our findings indicate that myonectin positively regulates muscle weight per body weight after denervation or DEX treatment. Furthermore, Ad-myonectin had no effects on body weight in SAMP8 and mdx mice compared with control Ad- β -gal

treatment. Our findings indicate that myonectin treatment increases muscle weight per body weight in SAMP8 and mdx mice.

“2) Muscle weight data and muscle cross-sectional area are interesting but incomplete. The authors should provide detailed histological analyses of the gastrocnemius and soleus muscles for the different models of muscle atrophy they utilized in Figures 1, 2, 6, and 7. Specifically, the authors should analyze the size distribution (gaussian plots) of different myofiber types, the percentage of myofiber types, and the total number of myofibers per muscle in soleus and gastrocnemius muscles from wild-type and myonectin knock-out mice in control conditions as well as when challenged by aging (Fig. 1), and by denervation and dexamethasone treatment (Fig. 2). Likewise, the analysis of myofiber size, type, and number of the gastrocnemius and soleus muscles should also be done for studies in Figures 6 and 7 that have analyzed the impact of recombinant/transgenic myonectin on muscle atrophy induced by denervation, senescence (SAMP8), and muscular dystrophy (mdx mice).”

Response: We thank the reviewer for these important suggestions. According to the reviewer's suggestion, we performed substantial histological analyses, such as myofiber size, myofiber type and myofiber number in muscle atrophy models. We found that myonectin KO mice had smaller cross sectional area (CSA) of type II muscle fibers in various models of muscle atrophy compared with WT mice. In contrast, CSA of type I muscle fibers did not differ in muscle atrophy models between WT and myonectin KO mice. Furthermore, myonectin treatment increased CSA of type II fibers, but not type I fibers in muscle atrophy models. These findings are presented in Result section and included as new Figures 1c, 2b, 2d, 6b, 7b and 7d, and new Supplementary Figures 3, 4b, 5, 6, 8 and 13 in the revised manuscript.

“3) Many of the in vivo studies (such as in Fig. 1) have utilized a low number of animals (n=4). This number of animals is insufficient especially for muscle functional assays (Fig. 1E), which have a high intrinsic variability. In figures 2, 6, and 7 more than 4 animals have been utilized in A but only muscles from n=4 were analyzed in B. It would be good if the authors could analyze the muscles from all the animals utilized.”

Response: We thank the reviewer for the important comments. We increased the number of animals in the case of the low number of animals (n=4). On the other hand, as the reviewer suggested, it would be good if the same number of animals are utilized in all analyses. However, each muscle sample is too small for all analyses, such as histological analyses, quantitative PCR analyses, Western blot analyses and electronic microscopical analyses. Thus, we examined some sets of in vivo analyses to perform all experiments.

“4) The authors suggest that myonectin prevents muscle wasting via activation of P-AMPK and PGC1a. However, they experimentally test the relevance of only AMPK downstream of myonectin. Notably, AMPK activation has been previously found to induce myofiber atrophy rather than protecting from it, and these effects have been ascribed to the capacity of AMPK to reduce mTOR activity and to induce autophagy. Therefore it remains unclear how myonectin protects from wasting via AMPK activation.”

“5) The PGC1a4 isoform (which induces hypertrophy) could provide a possible link between myonectin, AMPK, and protection from wasting but this link is not explored beyond the qPCR in Fig. 3B. In particular, the western blots have monitored general

PGC1a levels (rather than levels of the PGC1a4 isoform) and no rescue experiment has tested the functional interaction between myonectin and PGC1a4. How myonectin-AMPK would specifically regulate PGC1a4 also remains unclear.”

Response: We agree. As the reviewer pointed out, PGC1 α 4 isoform could induce muscle hypertrophy. Thus, we evaluated the protein levels of PGC1 α 4 in denervated muscle of mice and C2C12 myotubes. Based on the Reviewer #1's suggestion, we also assessed protein levels of IGF-1, which is upregulated by PGC1 α 4. The protein levels of PGC1 α 4 and IGF-1 in denervated skeletal muscle were significantly lower in myonectin KO mice than in WT mice (new Figure 3c). Treatment of WT mice with myonectin protein increased the protein levels of PGC1 α 4 and IGF-1 in denervated skeletal muscle (new Figures 6c and 6d). Furthermore, treatment of C2C12 myotubes with myonectin increased the protein levels of PGC1 α 4 and IGF-1 in the presence or absence of DEX (new Figure 5b). Therefore, these data suggest that myonectin could reduce muscle atrophy through upregulation of PGC1 α 4 and IGF-1 in skeletal muscle. These findings are presented in Result section and included as new Figures 3c, 5b, 6c and 6d in the revised manuscript.

We also dissected the mechanism of how myonectin-AMPK could regulate PGC1 α 4 expression. In C2C12 myotubes, myonectin-stimulated increase in PGC1 α 4 expression was canceled by siRNA-mediated knockdown of AMPK α 2 but not knockdown of AMPK α 1 (new Supplementary Figure 9). In addition, the stimulatory effects of myonectin on PGC1 α 4 protein level in skeletal muscle after denervation were not observed in dominant negative mutant form of AMPK transgenic mice (new Figure 6d). Thus, it is likely that myonectin increases PGC1 α 4 protein level in skeletal muscle via activation of AMPK α 2. These findings are presented in Result section and included as new Supplementary Figure 9 and new Figure 6d in the revised manuscript.

Moreover, we tested the functional interaction between myonectin and PGC1 α 4. Treatment of C2C12 myotubes with siRNA targeting PGC1 α reduced the expression of both PGC1 α and PGC1 α 4 compared with control siRNA treatment (new Supplementary Figure 10a). Knockdown of PGC1 α canceled myonectin-mediated increase in myotube diameter in the presence of DEX (Figure 5f). Knockdown of PGC1 α also reversed myonectin-stimulated increase in IGF1 expression in C2C12 myotubes in the presence of DEX (new Supplementary Figure 10b). Furthermore, overexpression of PGC1 α 4 by adenoviral vector system reversed the suppressive effects of AMPK inactivation on myonectin-induced increase in myotube diameter in the presence of DEX (new Supplementary Figure 11). These data indicate that myonectin could protect against DEX-induced myotube atrophy through the AMPK α 2/PGC1 α 4-dependent pathway. These findings are presented in Result section and included as new Supplemental Figures 10 and 11, and Figure 5f in the revised manuscript.

Taken together, our findings suggest that myonectin could prevent muscle atrophy through the AMPK α 2/PGC1 α 4/IGF-1-dependent mechanisms. These points are addressed in Discussion section in the revised manuscript.

“6) The authors report that myonectin mRNA levels decline with aging in soleus and gastrocnemius muscles (Fig 1A). Do myonectin mRNA levels change also in the other models of atrophy here utilized (denervation, dexamethasone treatment, and SAMP8 and MDX mice)? Because anti-myonectin antibodies are available, the authors could also explore whether myonectin protein levels change consistently with mRNA levels.”

Response: We thank the reviewer for these valuable suggestions. The protein levels of myonectin were reduced in all models of muscle atrophy, such as aging, denervation, dexamethasone treatment, SAMP8 and mdx mice. These findings are addressed in Result and Discussion section and included as new Supplementary Figures 1, 4, 16 and 17 in the revised manuscript.

“7) Muscle force has been assessed only for myonectin KO mice with aging (Fig. 1) but not for the other models: could the authors include muscle functional data also for dexamethasone-induced atrophy (WT versus myonectin KO mice) as well as for SAMP8 and MDX mice?”

Response: We thank the reviewer for the important suggestion. Maximal force of grip strength in 4 limbs and fore 2 limbs was significantly reduced in dexamethasone (DEX)-treated myonectin KO mice compared with DEX-treated WT mice.

As to SAMP8 and MDX mice, adenoviral vectors expressing myonectin or control vector were intramuscularly injected only to “right” gastrocnemius muscle of mice. Thus, the grip strength of 2limbs or 4limbs does not associate with effect of myonectin on muscle strength. Therefore, we evaluated muscle functional data only in DEX-induced atrophy model. These findings are now included as new Supplementary Figure 2e in the revised manuscript.

“8) The assays in Fig. 4C-D (analysis of citrate synthase activity and mitochondrial complex abundance) and Fig. 6C lack the WT and KO control from non-denervated animals. Moreover, there is no loading control for the blot in Fig. 4D.”

Response: We thank the reviewer for the kind suggestion. According to the reviewer's suggestion, we performed analysis of citrate synthase activity and mitochondrial complex abundance in gastrocnemius muscle from non-denervated WT and KO mice as well as denervated WT and KO mice (new Figures 4c and 4d). Expression of COX4 as a loading control was also assessed (new Figure 4c). In addition, we examined the protein levels of phosphorylated AMPK, AMPK, PGC1 α , PGC1 α 4 and tubulin of non-denervated WT and myonectin-KO mice, as well as denervated WT and myonectin-KO mice (new Figure 6c). These findings are now included as new Figures 4c, 4d and 6c in the revised manuscript.

Other points:

“9) Fibrosis is an important component especially of sarcopenia and muscular dystrophy therefore it would be interesting if the authors could test whether myonectin impacts the extent of fibrosis.”

Response: We thank the reviewer for the important suggestion. We evaluated interstitial fibrosis of gastrocnemius muscle in sciatic nerve denervation-induced muscle atrophy model by Masson Trichrome staining. Little fibrosis was observed in non-denervated or denervated gastrocnemius muscle in WT and myonectin-KO mice, and no differences were observed in the area of interstitial fibrosis between WT and myonectin-KO mice. These findings are presented in Result section and now shown in new Supplementary Figure 7 in the revised manuscript.

“10) The extent of body weight loss induced by dexamethasone treatment (Fig. 2) should be shown to determine whether myonectin knockout mice undergo similar whole-body wasting compared to controls in response to treatment with this drug.”

Response: We thank the reviewer for the important suggestion. As the reviewer pointed out, DEX treatment reduced whole body weight of WT and myonectin-KO mice compared with vehicle-treated WT and KO mice. On the other hand, there was no significant difference in body weight between WT mice and myonectin-KO mice after DEX treatment. These data are included as new Supplementary Figure 2d in the revised manuscript.

“11) The RNA-seq studies in Figure 3A should be reported in a more comprehensive manner apart the few examples of genes shown in Fig. 3B-C. It would be informative to report the gene categories regulated by myonectin.”

Response: We thank the reviewer for the useful suggestions. According to reviewer's suggestions, we further carried out Kyoto Encyclopedia of Genes and Genomes (KEGG) pathway enrichment analysis. We found that alternations of gene expression by myonectin were significantly associated with AMPK signaling pathway. These findings are presented in Result section and included in new Figure 3a in the revised manuscript.

“12) The studies with SAMP8 mice (Fig. 7A-C) are based on the transgenic delivery of myonectin versus control beta-Gal in 33-week-old mice and their subsequent analyses 4 months later. As additional control, the authors have examined animals before transgenic expression (33-week-old). It seems that only control animals transfected with beta-Gal differ from the control mice before treatment. Therefore, it is unclear whether there is an effect of myonectin or rather a detrimental effect of the beta-Gal control.

Do the authors have an untreated control of the same age of the animals that have been sacrificed (37-weeks)? If so, is there a difference in muscle wasting when comparing mice with transgenic myonectin expression versus untreated controls at the same age?”

Response: We thank the reviewer for the important suggestion. Muscle weights (normalized by body weights) of SAMP8 mice are as follows: untreated group at 33-week-old (before adenoviral vector (Ad) treatment); 4.18 ± 0.21 , untreated group at 37-week-old; 3.77 ± 0.05 , Ad- β -gal-treated group at 37-week-old; 3.85 ± 0.07 and Ad-myonectin-treated group at 37-week-old; 4.17 ± 0.06 . Thus, muscle weights of SAMP8 mice decline from 33-week-old to 37-week-old in agreement with previous report ¹. Muscle weights did not differ between untreated and Ad- β -gal-treated SAMP8 mice, indicating no detrimental effect of Ad- β -gal on muscle weight. Ad-myonectin increased muscle weights of SAMP8 mice compared with untreated control and Ad- β -gal control. Thus, myonectin can maintain the progression of muscle atrophy in SAMP8 mice.

“13) Fig. 7D-F: MDX mice are characterized by cycles of damage and regeneration. It is unclear how myonectin would be impacting these pathogenic processes in MDX mice. A detailed analysis of muscle regeneration would be needed if the authors want to report in a meaningful way the impact of myonectin on these models, including monitoring regeneration/myogenesis markers such as embryonic MHC and the number of centrally-nucleated myofibers.”

Response: We thank the reviewer for the important suggestion. Treatment of mdx mice with Ad-myonectin led to increased embryonic myosin heavy chain expression and

frequency of central nuclei-positive myocytes in gastrocnemius muscle tissue. These findings are presented in Result section and now included as new Supplementary Figures 17c and 17d in the revised manuscript.

Minor points:

“14) The H&E images of muscle cross-sections are of low resolution and hence not informative: they should be substituted with high-resolution histological images.”

Response: We thank the reviewer for the important suggestion. We changed all the photos of H&E staining from low resolution photos to higher resolution photos in the revised manuscript.

“15) The EM images in Fig. 4B are at low resolution and/or of insufficient contrast to discern ultrastructural defects in mitochondria – better images should be included.”

Response: We thank the reviewer for the important suggestion. We changed the electric microscope photo of non-denervated myonectin-KO mice (x5000) from low resolution photos to higher resolution photos in the revised manuscript (new Figure 4b).

“16) The original paper on PGC1a4 should be cited rather than a commentary on that paper.”

Response: We thank the reviewer for the kind suggestion. The original paper on PGC1a4² is now cited in the revised manuscript.

“17) The word “strengthen” has been used rather than “strength” and this should be corrected.”

Response: We thank the reviewer for carefully reading our manuscript. We now used the word “strength” instead of “strengthen” in the revised manuscript.

“18) There is no source file reporting the primary data. From the “source file” link, I can download only another copy of the same file that can be downloaded from the “supplementary dataset” link. Likewise, the full RNA-seq dataset reported in Fig. 3 should be made publicly available.”

Response: We thank the reviewer for the important suggestion. We deposited full RNA-seq data of Figure 3a in the Gene Expression Omnibus (Accession number: GSE233328). Link for reviewer only is attached below.

<https://www.ncbi.nlm.nih.gov/geo/query/acc.cgi?acc=GSE233328>

Private access password is “etoxiwyqpxczpgj”.

Reference

1. Guo AY, *et al.* Muscle mass, structural and functional investigations of senescence-accelerated mouse P8 (SAMP8). *Exp Anim* **64**, 425–433 (2015).

2. Ruas JL, *et al.* A PGC-1alpha isoform induced by resistance training regulates skeletal muscle hypertrophy. *Cell* **151**, 1319–1331 (2012).

REVIEWERS' COMMENTS

Reviewer #1 (Remarks to the Author):

The authors have addressed all of this reviewer's concerns and have provided new supporting data.

Reviewer #2 (Remarks to the Author):

The authors have done a good job at revising the manuscript.